# Medical Interpretability and Knowledge Maps of Large Language Models

**Razvan Marinescu**
Lumos AI, UCSC
razvan@thelumos.ai

**Victoria-Elisabeth Gruber**
Lumos AI
victoria@thelumos.ai

**Diego Fajardo**
Lumos AI
diego@thelumos.ai

## Abstract

We present a systematic study of medical-domain interpretability in Large Language Models (LLMs). We study how the LLMs both represent and process medical knowledge through four different interpretability techniques: (1) UMAP projections of intermediate activations, (2) gradient-based saliency with respect to the model weights, (3) layer lesioning/removal and (4) activation patching. We present knowledge maps of five LLMs which show, at a coarse-resolution, where knowledge about patient's ages, medical symptoms, diseases and drugs is stored in the models. In particular for Llama3.3-70B, we find that most medical knowledge is processed in the first half of the model's layers. In addition, we find several interesting phenomena: (i) age is often encoded in a non-linear and sometimes discontinuous manner at intermediate layers in the models, (ii) the disease progression representation is non-monotonic and circular at certain layers of the model, (iii) in Llama3.3-70B, drugs cluster better by medical specialty rather than mechanism of action, especially for Llama3.3-70B and (iv) Gemma3-27B and MedGemma-27B have activations that collapse at intermediate layers but recover by the final layers. These results can guide future research on fine-tuning, unlearning or de-biasing LLMs for medical tasks by suggesting at which layers in the model these techniques should be applied. Source code is available at https://github.com/TheLumos/medical-interpretability-llms.

## 1 Introduction

Large Language Models (LLMs) have achieved impressive performance on a variety of tasks, including for coding, reasoning and in-context learning Radford et al. (2019); Coignion et al. (2024); Hao et al. (2024); Wang et al. (2023); Szymanski et al. (2025); Nguyen et al. (2024). However, the understanding of the underlying mechanisms of how such models represent and store knowledge is still scarce. This is particularly important for medical tasks, where insights about the LLMs' representations of patient demographics, diseases and drug treatments can help uncover hidden biases, thus making such insights crucial for building safe and trustworthy models.

Over the past few years, a variety of mechanistic interpretability studies have been done on LLMs. A line of research has posited that all features learned by LLMs are linear, a claim known in the field as the *linear representation hypothesis* Bricken et al. (2023); Gurnee & Tegmark (2023); Heinzerling & Inui (2024). However, more recent work has shown that this hypothesis is not always true, and that LLMs can learn non-linear representations of features Engels et al. (2024), such as "days of the week" or "years of a century", which have circular patterns. Other lines of research have focused on understanding actual modules inside the LLM, such as the attention heads (AHs) Zheng et al. (2024). Thus, attention heads have been found that are specialized in associative memories Bietti et al. (2023), that identify particular structures e.g. the relation between the current token and the previous token Olsson et al. (2022); Nanda et al. (2023) or duplicate words Wang et al. (2022), that perform semantic processing related to the subject of the sentence Chughtai et al. (2024), that gather context Jin et al. (2024), that perform induction Edelman et al. (2024); Singh et al. (2024); Li et al. (2024); Crosbie & Shutova (2024) as well as other tasks related to latent reasoning and answer preparation, such as ensuring that the answers are correct Wiegreffe et al. (2024), coherent Guo et al. (2024) and faithful to the instructions Tanneru et al. (2024).

While most of interpretability studies have focused on general knowledge and tasks, few works have so far focused on understanding how LLMs represent and process medical knowledge. Some recent studies Chen et al. (2024); Savage et al. (2024) approached medical interpretability by asking the model to explain it's diagnosis decisions. He et al. has identified "modular circuits" in a medical-domain LLM (MedLlama-8B), which found certain circuits detect *patient symptoms* regardless of whether the model is doing diagnosis, treatment recommendation, or summarization. Wu et al. (2024) replaced the dense latent vectors of Llama 3 OpenBioLLM-70B with a sparse dictionary of medical features, where each dimension explicitly cooresponds to a medical concept. Kraišniković et al. (2025) fine-tuned BERT on a corpus of German pathology reports, and then projected the model's intermediate activations using t-SNE to visualize clusters of tumor pathologies. However, none of these studies systematically investigated LLM representations across a variety of medical knowledge areas, and in addition each study relied on a single explainability technique. In addition, each study only considered a single LLM. It is thus difficult to draw firm conclusions about the results and whether common representations might emerge across a variety of LLMs.

In this work we present a systematic interpretability analysis of five open-source LLMs (Llama3.3-70B, Gemma3-27B, MedGemma-27B, Qwen-32B and GPT-OSS-120B) for medical tasks. We study how the LLMs store knowledge about patient demographics such as age, as well as medical concepts such as symptoms, diseases, disease progression, drug treatments and drug dosages. We build LLM maps that visualize the layers where such knowledge is stored in the LLM. These maps are built by integrating four independent interpretability methods: (1) UMAP projections of intermediate activations, (2) gradient-based saliency with respect to the model weights, (3) layer lesioning (i.e. removing a layer and measuring the degradation in the response) and (4) activation patching (i.e. replacing the outputs of a single layer with the outputs obtained from a different prompt). We choose these techniques due to their simplicity as well as their signficant previous use in the literatureMcInnes et al. (2018a); Gana et al. (2024); Zhang et al. (2025); Dai et al. (2021). In addition, we find several interesting phenomena: (i) age is encoded in a non-linear manner at intermediate layers in the models, and sometimes shows discontinuities, such as between subjects younger than 18 and those older than 18, (ii) disease progression is circular and hence non-monotonic at certain layers of the model, where late-stage embeddings come closer to the early-stage embeddings (iii) Llama3.3-70B internally has a drug representation that aligns better with medical specialty rather than mechanism of action, (iv) Gemma and MedGemma models have activations that collapse at intermediate layers. The prompts and the medical/drug classifications were verified by a medical professional (Clinical Neuroscience PhD) on our team.

## 1.1 RELATED WORK

**UMAP Embeddings:** Zhang et al. (2025) have used UMAP embeddings to visualize intermediate activations in an LLM trained on single-cell RNA data. Bolukbasi et al. (2021); Reif et al. (2019) performed interpretability studies using UMAP on BERT. Yeh et al. (2023) built Attentionviz, a tool for visualizing intermediate embeddings in attention-based models generated using UMAP/t-SNE/PCA.

**Weight Saliency:** Dai et al. (2021) used a gradient-based saliency method to identify knowledge neurons in pretrained transformers. Fan et al. (2023) used gradient-based weight saliency to un-learn the generation of harmful images. Frankle & Carbin (2018) used weight sparsity, a form of weight saliency, to achieve better generalization in neural network predictions.

**Layer and Attention Head Lesioning:** In neuroscience, lesioning studies, which involve removing a brain region and studying which cognitive, motor or sensory functions are lost, have successfully identified functions of specific brain regions. In AI interpretability, Logit-Lens nostalgebraist and Tuned-Lens Belrose et al. (2023) were used to interpret models such as GPT-Neo-2.7B by taking intermediate activations at a particular layer and computing the logit directly with it, thus removing all layers after it. Michel et al. (2019) studied BERT outputs after removed one attention head at a time, as well as all-but-one attention heads within a layer. Voita et al. (2019) used layer-wise relevance propagation Binder et al. (2016) to study the importance of different attention heads before pruning, and they found that only a few specialized heads do the heavy lifting. Zhang et al. (2024) performed a layer ablation study on Llama3-70B and two smaller LLMs and found that only a few layers were important, whereby removing them led to a collapse in the model's performance. Gromov et al. (2024) removed more and more layers in Llama2, Qwen, Mistral and Phi-2, followed

by a small amount of fine-tuning, and found that only by the time 50% of layers were removed did the models' performance significantly degrade.

**Causal interventions and Activation Patching:** Causal interventions such as Activation patching Heimersheim & Nanda (2024); Zhang & Nanda (2023) and Rank-One Model Editing (ROME) Meng et al. (2022) have been used to identify circuits in LLMs and transformer-based models, or even functions of specific activation heads Hex. Fierro et al. (2024) used activation patching to localize the role of language during knowledge recall.

## 2 METHODS

In Fig. 1 we show an overview of our Medical LLM Interpretability study, outlining the process to build LLM maps. We run the LLM through a variety of prompts related to medical knowledge: patient age, symptoms, diseases, drug treatments and drug doses. For each such knowledge area we run four different interpretability methods: (1) UMAP projections of intermediate-layer activations from which we extract quantitative metrics such as Silhuette (cluster separation) scores, (2) gradient-based saliency of model weights (3) layer lesioning, where we replace each layer with the identity function $I_n$ and score the degradation in the altered response, and (4) activation patching, where we replace the outputs of a single layer with the outputs obtained from a different prompt. We finally extract quantitative measures from each of these methods (top-right) and plot the LLM map consisting of the layer intervals where the highest quantitative measures are found. Since each method makes different assumptions and has different strengths and weaknesses, using all of them together helps give confidence that the layers identified are indeed the ones where the medical knowledge is stored in the LLM.

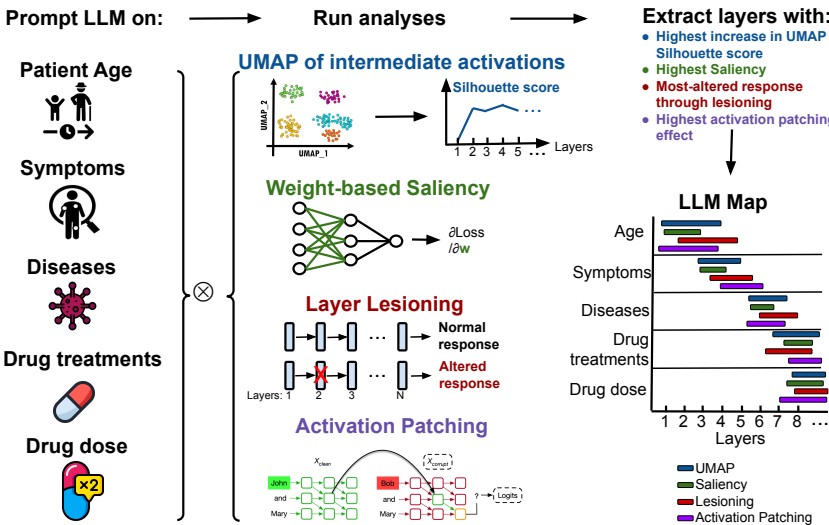

Figure 1: Overview of our Medical LLM Interpretability study, outlining the process to build LLM maps.

**UMAP:** We use the Uniform Manifold Approximation and Projection (UMAP) McInnes et al. (2018b) to project the intermediate activations of the LLM into a 2D space for visualization. We use $n\_neighbors = 15$, $min\_dist = 0.1$ and a cosine distance metric to build the UMAP connectivity graph. To get a quantitative measure of clustering, we compute the Silhuette Shahapure & Nicholas (2020) (cluster separation) score for each layer, which is a measure of how well the activations cluster in the lower-dimensional space. The Silhuette score is defined as:

$$s(i) = \frac{b(i) - a(i)}{\max(a(i), b(i))} \tag{1}$$

where $a(i)$ is the average distance between $i$ and all other points in the same cluster, and $b(i) = \min_{j \neq i} \frac{1}{n_j} \sum_{x_j \in C_j} d(x_i, x_j)$ is the minimum average distance between $i$ and points in the next closest cluster $C_j$. To get a per-layer value, we average the Silhouette score across all prompts. We chose Silhouette because it directly measures ground-truth label separability without requiring running a clustering algorithm such as K-means. For the Silhouette score, we project using UMAP on $D = 30$ dimensions (instead of $D = 2$ used for visualization) in order to retain more information about the original datapoints. We further compute confidence intervals for the Silhouette score using bootstrap resampling.

For age-related prompts, which query patients with ages from 1 to 100, we instead compute *local anisotropy*, a measure of "1D-ness" of the datapoints. Taking the 2D UMAP-embeddings, the local anisotropy of a single embedding $x_i$ is defined as $A_i = 1 - \frac{\lambda_2}{\lambda_1}$, where $\lambda_1 \geq \lambda_2$ are the largest two eigenvalues of the local covariance matrix computed from the $k = 20$ nearest neighbors of each point. The final anisotropy value is averaged across all points.

**Weight Saliency:** We compute the gradient-based saliency of the model weights for each layer by computing $\frac{\partial \log \mathcal{L}}{\partial w}$ where $\mathcal{L}$ is the loss function and $w$ are the model weights. To get a per-layer value, we average the saliency across all weights in the attention heads and MLP at each layer. To obtain more reliable results, we average these values across multiple prompts, and also use these statistics to compute confidence intervals.

**Layer Lesioning:** We replace each layer in the LLM with the identity function $I_n$ and score the degradation in the altered response. For scoring the degradation, we use an LLM-as-a-judge (here GPT-4o) and use a rubric where a score of 1/10 means no degradation compared to the original response and 10/10 means full degradation with gibberish responses.

**Activation Patching:** Activation patching is a technique that replaces select internal activations of a neural network model. It runs multiple forward passes: a clean run ($cl$), a corrupted run ($*$), and a patched run ($pt$). The patching effect is defined as the gap of the model performance between the corrupted and patched run, under an evaluation metric. For our study, we follow the best practices from Heimersheim & Nanda (2024); Zhang & Nanda (2023). In particular, we only patch the attention heads and the MLP blocks at each layer. We compute the patching effect using the normalized logit difference: $P = \frac{LD_{pt}(r,r') - LD_*(r,r')}{LD_{cl}(r,r') - LD_*(r,r')}$, where $LD(r,r') = \text{Logit}(r) - \text{Logit}(r')$ is the logit difference between responses $r$ and $r'$ (in our case, $r$ and $r'$ are single-token responses such as *Angina* and *Pneumonia* respectively). This normalized value typically lies in the range $[0,1]$, where 1 signifies fully restored performance after patching and 0 signifies patching has no effect. To obtain more reliable results, we average these values across multiple prompts (see Appendix sec. A), and also use these statistics to compute confidence intervals.

**LLM Maps:** In order to plot LLM Maps showing where medical knowledge is stored inside the LLM, we select continuous layer intervals from each of the four analyses as follows:

- For UMAP, for all prompt types except age, we select the layer interval showing the highest increase in Silhouette score after Gaussian smoothing ($\sigma = 1.0$), using a sliding window of 3 layers to find the maximum average rate of increase. For age-related prompts, we use local anisotropy instead of Silhouette score.

- For saliency, layer lesioning and activation patching, we select intervals where values are above the 75th percentile threshold after Gaussian smoothing, requiring minimum 2-layer intervals and taking up to 3 intervals per analysis.

We finally plot the LLM map consisting of the layer intervals where the highest quantitative measures are found. Since each method makes different assumptions and has different strengths and weaknesses, using all of them together helps give confidence that the layers identified are indeed the ones where the medical knowledge is stored in the LLM.

## 3 RESULTS

**LLM Medical Maps**: In Fig. 2 we show the LLM map for Llama3.3-70B showing where medical knowledge about age, symptoms, diseases, drugs and drug dosages is stored in the LLM. We find

that a subject's age is learned/processed in layers 0-5, medical symptoms are learned in layers 0-9 as well as 15-40. Knowledge about diseases seems to be found in layers 0-5 or potentially 27-37, while knowledge about drugs is most likely learned in layers 15-45. Finally, knowledge about drug dosage is likely learned in the first half of the layers (0-40), although the results seem more inconclusive. For Drug dosage, we could not use the UMAP results due to the lack of a clear quantitative measure that we could compute (see Appendix Fig. 22 for an example plot of UMAP results for drug dosage.)

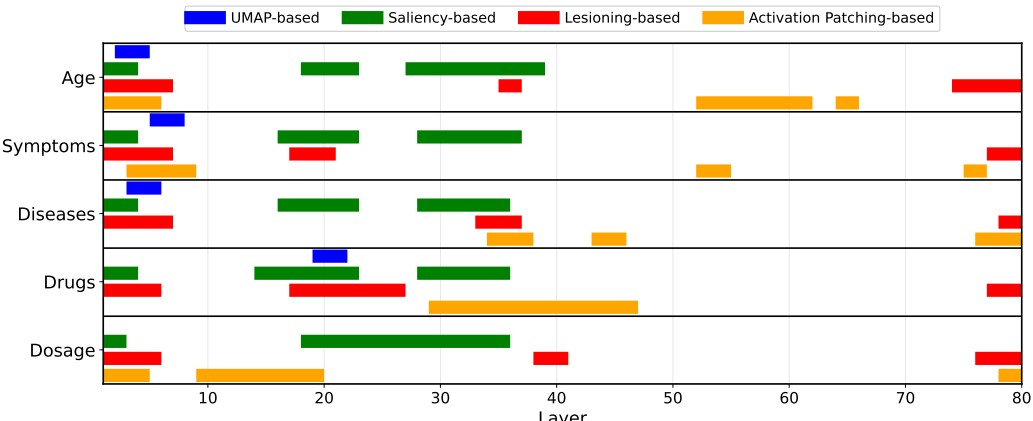

Figure 2: LLM Map for Llama3.3-70B showing where medical knowledge about age, symptoms, diseases, drugs and drug dosages is stored. Each interval shown is estimated quantitatively from four analyses: (1) clutering structure in UMAP embeddings, (2) high weight saliency values, (3) high degradation upon layer lesioning/ablation and (4) high patching effect in activation patching.

**Age manifold is non-linear**: In Fig. 3 we show the UMAP analysis for Llama3.3-70B using prompts about subjects with different ages, broken down by gender. The age manifold is gender-dependent, and further shows non-linearities throughout many intermediate layers, and sometimes altogether a complete change of direction of age progression. In the last plots we plot the true vs predicted age from a linear regression model at layer 37, where the largest discontinuity is found for "she" prompts. Since the discontinuity is between ages 17 and 18, it likely captures a separation between teenagers and adults that the model has learned. A similar age discontinuity between 17 and 18 years is found for "he" and "someone" prompts (results not shown) at the same layer.

**Disease Progression representation is circular and non-monotonic**: In Fig. 4 we show UMAP-projected intermediate layer activations for Llama3.3-70B using prompts describing the progression of four different diseases: Alzheimer's disease, Parkinson's, COVID-19 and COPD. The disease progression manifold shows multiple instances of "circular patterns" across all diseases analyzed. By circular here we mean that the prompts describing late disease stages come back to a closer location to the early-stage prompts. In the penultimate sub-plot we show the closest stages to the first stage (completely healthy with no symptoms), ignoring stage 2, and we find that for many diseases, in particular Parkinson's and COPD, late stages (5-9) are closer to stage 1 than earlier stages (e.g. 3-4), suggesting that the disease progression is non-monotonic. A similar result is confirmed in the final subplot, where for Alzheimer's disease in particular stages 2-4 are closer to the end stage (death) than later stages in the representations at many intermediate layers.

**Drugs cluster both by mechanism and specialty**: In Fig. 5 we show UMAP-projected intermediate layer activations for Llama3.3-70B using prompts about different drugs, broken down by mechanism of action. In Fig. 6, we show similar results for drugs broken down by medical specialty. The LLM learns to represent drugs both by mechanism of action and by medical specialty. Quantitative comparisons with Silhouette Scores (Table 1) confirm that across most models, drugs cluster more by medical specialty than mechanism of action.

**For Gemma/MedGemma, activations collapse at intermediate layers**: In Fig. 7 we show UMAP-projected intermediate layer activations for Gemma-27B. We notice that the activations collapse in UMAP space at certain intermediate layers (here layer 20 is shown), although they recover in later layers. Results with more intermediate layers are shown in the Appendix section D. We found similar results for MedGemma-27B (see Appendix Fig. 19).

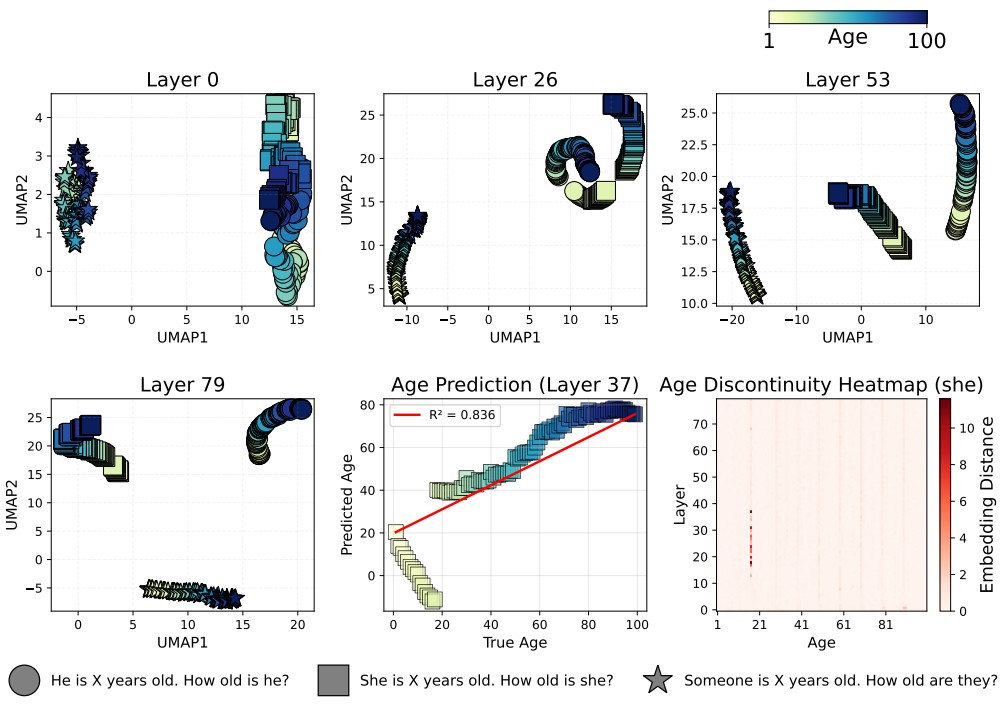

Figure 3: UMAP Analysis using subjects with different ages in Llama3.3-70B. The age manifold shows non-linearities throughout many intermediate layers, as well as a discontinuity between subjects younger than 17 and those 18 or older. Prompts used are shown at the bottom of the figure. The bottom-right plot confirms that the only discontinuity is at age 18.

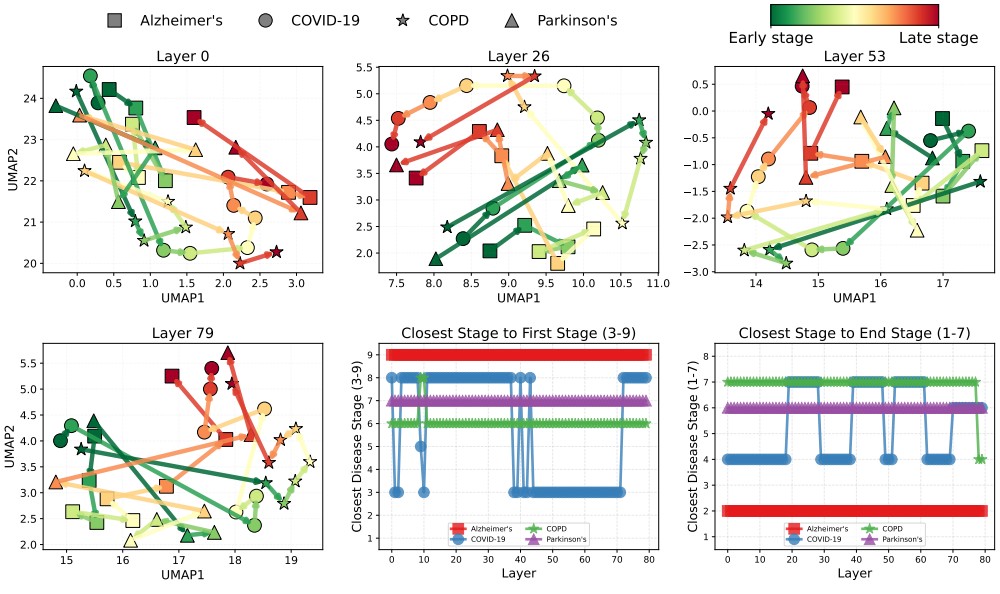

Figure 4: Disease Progression UMAP in Llama3.3-70B. The model shows circular, non-monotonic disease progression, in particular for Alzheimer's disease (stages 2 and 4 are closest to the final stage at many layers) and Parkinson's disease (stage 7 is closest to first stage at most layers). Details about each stage as it was prompted in the model is shown in Appendix Table. 5.

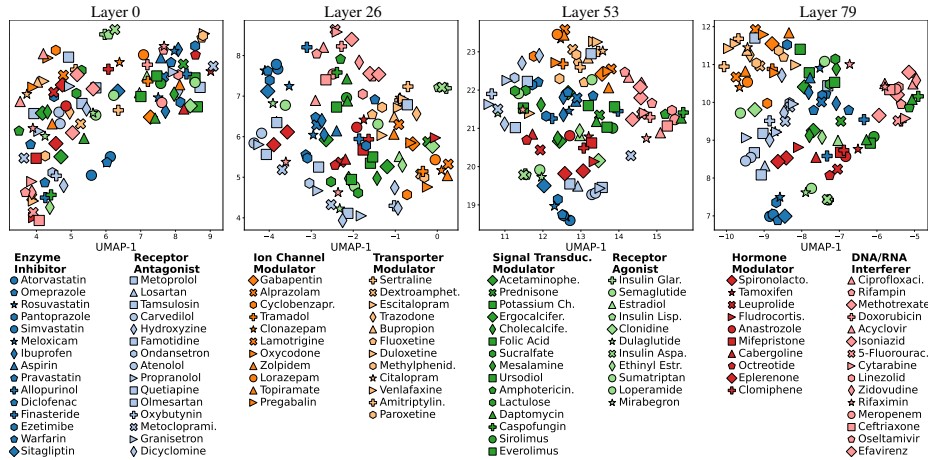

Figure 5: Drug UMAP embeddings colored by mechanism of action in Llama3.3-70B.

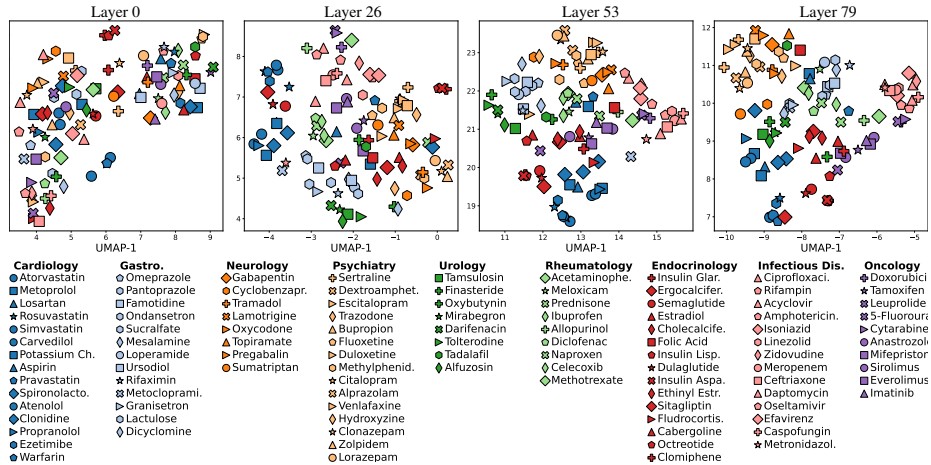

Figure 6: Drug UMAP embeddings colored by medical specialty in Llama3.3-70B

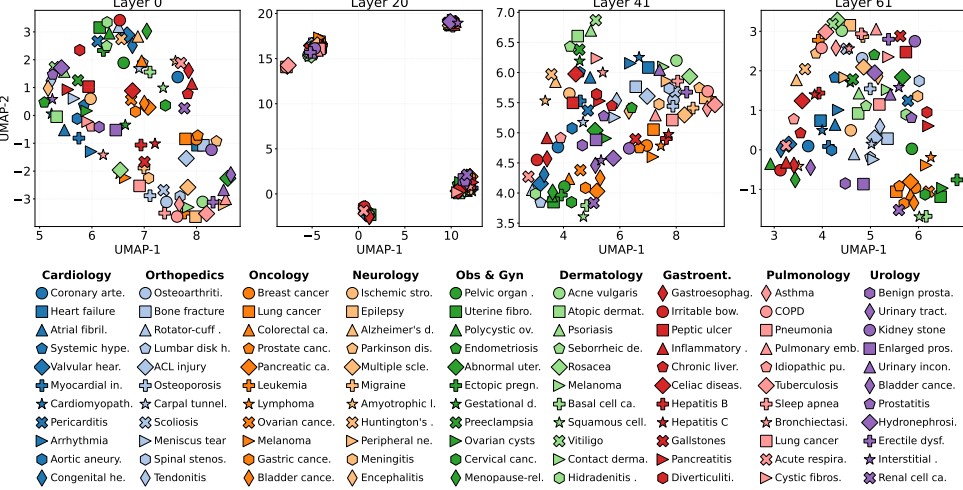

Figure 7: Intermediate layer activations collapse in UMAP space for Gemma (shown above), although they recover in later layers. Similar results are observed for MedGemma. Full results with all layers are shown in the appendix.

**Quantitative interpretability metrics**: Across all 5 LLM and all medical areas, we computed key interpretability metrics which we show along with the best layer where these are achieved (Table 1). For Age, we compute a measure of linearity as the $R^2$ of the linear fit between the true age and the age predicted from a linear fit on the embeddings in the 2D UMAP space. For Symptoms, Diseases and Drugs, we show the Sillhouette score, while for disease progression we computed two proxy measures of circularity: (1) the mean $\mu_{CSFS}$ of *closest stages to first stage* (CSFS), excluding stage 2 and (2) the mean $\mu_{CSLS}$ of *closest stages to end stage* (CSLS), excluding the penultimate stage. We excluded the second and penultimate stages due to it being affected by the granularity of the disease progression. For Dosages, we show the patching effect as obtained from activation patching. For all metrics, we show the scores obtained at the best layer. We also show the same metrics averaged across all layers (Table 2) and at the last layer (Table 6).

The results show that most models can achieve linear age manifolds at some intermediate layer, even though at the final layer Gemma3-27B and Qwen-32B only achieve an $R^2$ of around 0.6 (Table 6). The best scores for Silhouette scores are achieved mainly by Llama3.3-70B and Qwen-32B. GPT-OSS-120B also achieves good scores in a variety of categories. MedGemma-27B and Gemma3-27B consistently achieve low scores in all categories except Disease Progression at the final layer.

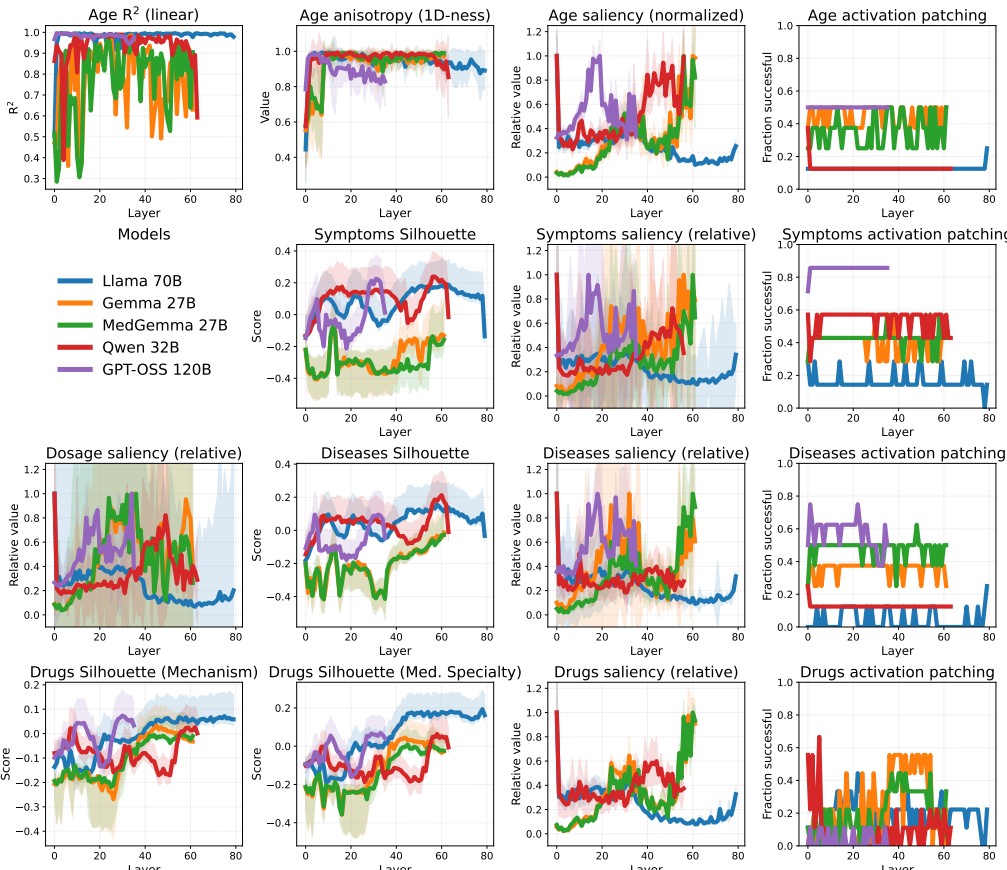

Figure 8: Selected interpretability metrics per layer for the five models. (first row) Age $R^2$ measuring linearity of age manifold, age anisotropy (a measure of 1D-ness), age gradient-based saliency and fraction of successful activation patches. (second row) medical symptoms metrics, (third row) drug dosage saliency and disease metrics and (fourth row) drug treatment metrics. Lines of some models terminate early due to having a smaller number of layers (e.g. GPT OSS has only 36 layers).

We further plot key interpretability metrics across all layers for each model (Fig. 8). Llama and GPT show the highest $R^2$ across layers. Llama and Qwen show the highest UMAP Silhouette scores for symptoms and diseases, while the last layers of Llama show highest UMAP Silhouette scores for drugs. Highest saliencies are observed in the last layers of Gemma/Medgemma, first layer of Qwen, and layers 15-22 of GPT-OSS. Activation patching analyses show that patching succeeded

(patching effect $> 0.5$) more often in GPT and Gemma/Medgemma than in the other models. Layer Lesioning analyses (Appendix Fig. 9) show that removing layers in Gemma/Medgemma seems to lead to severe degradation in the response, while the other models are most robust and thus have redundancy in their layers. Llama and GPT generally show the highest robustness to layer removal, for both early layers and late layers.

Table 1: Per-concept interpretability metrics across models. Values are best layer scores with the layer index. CSFS: Closest Stage to First Stage, CSLS: Closest Stage to Last Stage

| Concept | Metric | Llama 70B | | Gemma 27B | | MedGemma 27B | | Qwen 32B | | GPT-OSS 120B | |
|---|---|---|---|---|---|---|---|---|---|---|---|
| | | Score | Layer | Score | Layer | Score | Layer | Score | Layer | Score | Layer |
| Age | $R^2$ (linear) ↑ | **1.00 ± 0.00** | 3 | 0.99 ± 0.00 | 34 | 0.98 ± 0.01 | 24 | 0.99 ± 0.00 | 25 | 1.00 ± 0.00 | 1 |
| Symptoms | Silhouette ↑ | 0.19 ± 0.04 | 63 | -0.10 ± 0.06 | 12 | -0.08 ± 0.05 | 12 | **0.24 ± 0.05** | 56 | 0.23 ± 0.04 | 31 |
| Diseases | Silhouette ↑ | 0.16 ± 0.05 | 58 | -0.01 ± 0.04 | 61 | -0.03 ± 0.04 | 61 | **0.21 ± 0.04** | 60 | 0.10 ± 0.05 | 32 |
| Disease Progression | CSFS ↓ | 6.25 ± 2.17 | 0 | **4.25 ± 1.09** | 0 | **4.25 ± 1.09** | 0 | **4.25 ± 1.09** | 2 | 5.00 ± 1.58 | 3 |
| Disease Progression | CSLS ↑ | 5.50 ± 2.06 | 74 | 5.75 ± 1.30 | 54 | 5.75 ± 1.30 | 52 | 6.75 ± 0.43 | 44 | **6.75 ± 0.43** | 29 |
| Drugs | Silhouette (mech.) ↑ | 0.07 ± 0.08 | 75 | 0.03 ± 0.09 | 45 | -0.01 ± 0.08 | 44 | 0.03 ± 0.06 | 62 | **0.07 ± 0.07** | 30 |
| Drugs | Silhouette (spec.) ↑ | **0.19 ± 0.13** | 78 | 0.05 ± 0.12 | 44 | 0.01 ± 0.10 | 45 | 0.06 ± 0.07 | 58 | 0.07 ± 0.07 | 33 |
| Dosages | Patching Effect ↑ | 2.64 ± 1.36 | 79 | 0.92 ± 0.61 | 8 | 1.71 ± 1.49 | 0 | **7.33 ± 12.29** | 0 | 4.23 ± 11.04 | 23 |

Table 2: Per-concept interpretability metrics across models. Values are mean ± std across all layers.

| Concept | Metric | Llama 70B | Gemma 27B | MedGemma 27B | Qwen 32B | GPT-OSS 120B |
|---|---|---|---|---|---|---|
| Age | $R^2$ (linear) ↑ | 0.98 ± 0.06 | 0.78 ± 0.16 | 0.78 ± 0.18 | 0.93 ± 0.09 | **0.98 ± 0.01** |
| Symptoms | Silhouette ↑ | 0.08 ± 0.08 | -0.28 ± 0.09 | -0.30 ± 0.08 | **0.10 ± 0.09** | -0.02 ± 0.12 |
| Diseases | Silhouette ↑ | **0.05 ± 0.07** | -0.21 ± 0.11 | -0.22 ± 0.10 | 0.04 ± 0.08 | -0.06 ± 0.09 |
| Disease Progression | CSFS ↓ | 6.96 ± 1.77 | 5.90 ± 2.14 | **5.72 ± 2.13** | 7.21 ± 1.59 | 6.58 ± 1.88 |
| Disease Progression | CSLS ↑ | 5.08 ± 2.00 | 4.68 ± 2.10 | 4.66 ± 2.07 | 5.23 ± 1.98 | **5.65 ± 1.47** |
| Drugs | Silhouette (mech.) ↑ | **-0.01 ± 0.08** | -0.10 ± 0.09 | -0.10 ± 0.08 | -0.08 ± 0.06 | -0.02 ± 0.07 |
| Drugs | Silhouette (spec.) ↑ | **0.05 ± 0.13** | -0.12 ± 0.12 | -0.14 ± 0.10 | -0.09 ± 0.07 | -0.03 ± 0.07 |
| Dosages | Patching Effect ↑ | 2.20 ± 1.17 | 0.79 ± 0.70 | 1.25 ± 0.85 | **5.16 ± 7.84** | 2.16 ± 5.77 |

We also computed quantitative metrics on six other medical and life science LLMs: OpenBioLLM-70B, PMC-LLaMA-13B, ClinicalCamel-70B, PalmyraMed-70B, Meditron-70B, HuatuoGPT-70B. We show the metrics for the best layer in Table 3, averages across all layers in Supplementary Table 7 and the last layer metrics in Supplementary Table 8.

Table 3: Per-concept interpretability metrics across medical LLMs. Values are best layer scores (max across depth) with the layer index.

| Concept | Metric | Llama3 OpenBioLLM 70B | | PMC LLaMA 13B | | Clinical Camel 70B | | Palmyra Med 70B | | Meditron 70B | | HuatuoGPT o1 70B | |
|---|---|---|---|---|---|---|---|---|---|---|---|---|---|
| | | Score | Layer | Score | Layer | Score | Layer | Score | Layer | Score | Layer | Score | Layer |
| Age | $R^2$ (linear) ↑ | 1.00 ± 0.00 | 16 | 0.99 ± 0.00 | 5 | 0.99 ± 0.00 | 38 | **1.00 ± 0.00** | 3 | 0.99 ± 0.00 | 20 | 1.00 ± 0.00 | 3 |
| Symptoms | Silhouette ↑ | **0.22 ± 0.04** | 51 | 0.16 ± 0.04 | 30 | 0.15 ± 0.05 | 58 | 0.20 ± 0.04 | 47 | 0.16 ± 0.04 | 68 | 0.18 ± 0.04 | 50 |
| Diseases | Silhouette ↑ | 0.07 ± 0.04 | 61 | 0.12 ± 0.04 | 26 | 0.15 ± 0.04 | 10 | 0.16 ± 0.04 | 54 | **0.16 ± 0.05** | 52 | 0.11 ± 0.04 | 59 |
| Disease Progression | CSFS ↓ | 5.50 ± 2.60 | 0 | 4.25 ± 0.83 | 9 | **4.00 ± 0.71** | 19 | 5.75 ± 2.38 | 20 | 5.25 ± 1.48 | 19 | 6.25 ± 2.17 | 0 |
| Disease Progression | CSLS ↑ | **7.00 ± 0.00** | 34 | 6.50 ± 0.50 | 38 | 6.75 ± 0.43 | 64 | 5.50 ± 2.06 | 72 | 6.75 ± 0.43 | 71 | 5.50 ± 2.06 | 74 |
| Drugs | Silhouette (mech.) ↑ | 0.05 ± 0.06 | 48 | 0.09 ± 0.05 | 28 | **0.14 ± 0.07** | 50 | 0.09 ± 0.05 | 24 | 0.13 ± 0.07 | 51 | 0.11 ± 0.05 | 41 |
| Drugs | Silhouette (spec.) ↑ | 0.15 ± 0.10 | 62 | 0.13 ± 0.07 | 27 | 0.22 ± 0.10 | 47 | 0.23 ± 0.10 | 47 | 0.23 ± 0.11 | 45 | **0.26 ± 0.11** | 46 |
| Dosages | Patching Effect ↑ | 3.22 ± 1.20 | 79 | 16.00 ± 31.30 | 0 | **21.25 ± 46.75** | 1 | 0.26 ± 0.56 | 79 | 1.78 ± 3.32 | 49 | 1.99 ± 1.03 | 9 |

## 4 DISCUSSION

The knowledge map we derived for Llama3.3-70B shows that most medical knowledge is stored in the first half of model layers, while the later half of the model likely stores other types of knowledge. However, GPT-OSS-120B and Qwen-32B show an oppoosite effect, most analyses point to layers in the second half of the model (see appendix sec. C). This suggests that methods trying to finetune these models for medical tasks, or un-learn (Fan et al. (2023)) certain medical biases, should focus on the layers we identified. In addition, our work helps suggest where causal interventions on the models should be focused, in order to change medical concepts.

The non-linearities and discontinuities of the age manifold suggest that age is not learned as a disentangled dimension in the LLM model. This will make age-based re-learning or de-biasing more difficult. In addition, the discontinuities noticed at age 18 suggest that Llama might consider the young subjects in a group of their own, and might display further biases towards that group of young people in a variety of dimensions. Further investigations are needed to understand exactly what these biases could be.

The circularity of the disease progression manifold is somewhat suprising. Previous research by Engels et al. (2024) has found other types of internal representations that are circular, such as the days of the week of years in a century. While we are not certain about the benefits of such a representation for medical tasks, we hypothesize that a useful representations for disease progression would have three desiderata: (1) it would contain a common healthy state across all diseases, (2) optionally a common death state across all diseases and (3) for certain diseases, the disease progression trajectories would converge at key mid-points; for example, in Alzheimer's disease, it is hypothesized that both amyloid as well as tau pathology eventually lead to neurodegeneration and brain atrophy, thus the state of neuro-degeneration could be a converging point for both trajectories, eventually leading to advanced cognitive impairment and dementia. While Llama did satisfy (1) and (2) according to our prompts, for (3) we did not find such converging points in our study. However, due to the difficulty of creating suitable prompts, we only did this for a limited number of diseases, so further work needs to be done in order to validate this.

The collapse of intermediate layer activations for Gemma/MedGemma models warrant further investigation. Even though the activations do recover, we believe there is a possibility that representational power is being wasted in those layers where representations collapse. Further investigations are needed to understand the nature of the collapse, whether it is caused by the model's architecture, the training regime, or the data used for training, and the implications of such a collapse.

### 4.1 ACTIONABLE RECOMMENDATIONS

**Removing age discontinuities and enforcing linear manifold for ages:** If one wants to de-bias the age in order to remove the discontinuity at 18 years or to make the age manifold entirely linear, one can fine-tune only those exact layers where age is processed and use an age regularizer in the loss function that maintains linearity, such as $\mathcal{L}_{\text{age}} = \lambda * (1 - \text{R}(\text{ages}, \text{UMAP}(\text{prompt}(\text{ages}))))$, where R is the linear regression coefficient which will be 1 if the manifold of age embeddings is entirely linear.

**Preventing collapse of Gemma/MedGemma:** To prevent the collapse of Gemma/MedGemma, the Google team can add during training a Uniformity loss on a hypersphere Wang & Isola (2020) as follows: $\mathcal{L}_{\text{unif}} = log \, \mathbb{E}_{i \neq j} \left[ \exp(-\alpha ||z_i - z_j||^2) \right]$. A large $\alpha$ will incur a large penalty for small distances between embeddings $z_i$ and $z_j$, thus preventing the collapse into 3-4 blobs as seen in the UMAP Fig 7.

**Enforcing monotonicity in disease progression:** During training, AI labs can add a regularizer that enforces monotonic disease progression, such as:

$$\mathcal{R}_{\text{monotonic}} = \lambda \sum_{s=1}^{S-1} \left[ \max\left(0, \, d(h_s, h_1) - d(h_{s+1}, h_1)\right) \right]^2$$

which enforces the hidden-state activations $h_s$ at disease stage $s$ to move further away from $h_1$, the activation for stage 1 (healthy), where $d(\cdot, \cdot)$ is a chosen distance function. These regularizerscan be made even more complex by anchoring not just at $h_1$, but at other intermediate stages as well.

**Limitations:** With lack of ground-truth data, it is difficult to validate the results we derived. In order to alleviate this, we ran four different analyses (UMAP, saliency, layer lesioning and activation patching) on the same model and thus allow the reader to see the consistency of the results across different methods. Since the methods operate with entirely different assumptions, where they are in agreement, we can be significantly more confident in the results we derived. Even when they are not in agreement, some conclusions can still be drawn. For example in the case of drug dosage, both saliency and activation patching indicate signal in the first half of the model layers (even though they disagree on the exact layers), so we can at least draw some broader conclusions even in this case. In addition, as opposed to other interpretability studies, we note that we cannot use human experts to provide ground-truth data due to the lack of understanding of the model's internal representations.

### ACKNOWLEDGMENTS

This study was supported by Lumos AI. Computational credits were provided by Google Cloud.

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

APPENDIX

**Notes on LLM usage:** We used LLMs for the following:

- For brainstroming ideas, such as on how to implement the hooks for activation patching and lesioning
- For generating portions of our code, such as specific functions, and for debugging
- For auto-completions during the writing of the latex file, mainly for latex code generation for tables and figures
- For generating some of the prompts used in the analyses, which we then manually reviewed and edited

## A  PROMPTS USED

Table 4: Prompt templates used across all analyses. Blue words are automatically substituted with a variety of values in order to test multiple ages, symptoms, diseases, drugs and drug dosages. Green words denote the expected answers with respect to which we compute gradients (saliency) or logit differences (activation patching).

| Analysis | Prompt Templates |
|---|---|
| **Age** | |
| UMAP | `He is {20} years old.` |
| Saliency | `Answer with a single number, nothing else.  At what age is someone legally considered an adult? {18}` |
| Lesioning | `How long do people live around the world?` |
| Activation Patching | `Clean:A {75}-year-old patient has {memory problems}.  The most likely cause is ___{Dementia}`
`Corrupt:A {5}-year-old patient has {memory problems}.  The most likely cause is ___{Epilepsy}` |
| **Symptoms** | |
| UMAP | `A patient has {headache}.` |
| Saliency | `Answer with a single word.  What is the main symptom present in {Pneumonia}? {headache}` |
| Lesioning | `A patient has a lupus.  What symptoms are associated with this disease?` |
| Activation Patching | `Clean:For {migraine}, a common symptom is ___{headache}`
`Corrupt:For {epilepsy}, a common symptom is ___{seizures}` |
| **Diseases** | |
| UMAP | `A patient was diagnosed with {COPD}.` |
| Saliency | `A patient has fever, productive cough, and chest crackles on exam.  The most likely diagnosis is _{Pneumonia}` |
| Lesioning | `A 30-year-old female patient presents with chronic cough, shortness of breath, and wheezing.  What is the most likely diagnosis?` |
| Activation Patching | `Clean:A patient has {chest pain and shortness of breath}.  The most likely diagnosis is ___{Angina}`
`Corrupt:A patient has {chest pain and coughing}.  The most likely diagnosis is ___{Pneumonia}` |
| **Drugs** | |
| UMAP | `A patient takes {Atorvastatin}.` |
| Saliency | `Answer with a single word only, describing a drug.  A patient suffers from high LDL cholesterol and has atherosclerotic disease.  What drug should the patient take? {Atorvastatin}` |
| Lesioning | `A patient takes Metformin.  What are the side effects?` |
| Activation Patching | `Clean:The patient has {ADHD}.  A common medication is ___ {Adderall}`
`Corrupt:The patient has {depression}.  A common medication is ___{Sertraline}` |
| **Drug Dosage** | |
| UMAP | `A patient took {5}mg of {Amlodipine}.` |
| Saliency | `A patient has taken {5} mg of {Amlodipine}.  Will the patient be alive or dead? {alive},{dead}` |
| Lesioning | `What is a safe dose of Amlodipine and what is a lethal dose?` |
| Activation Patching | `Clean:What will happen to the patient?  Answer in exactly one English word:  stable or dead.  A patient took {5}mg of {Amlodipine}.  The patient will be _{stable}`
`Corrupt:Same as the clean prompt, except the dose changed to {100}mg, and expected answer changed to {dead}` |

Table 5: Disease progression prompts used for UMAP analysis. Each disease has 9 stages from healthy to death, showing realistic progression patterns.

| Disease | Progression Prompts |
|---------|---------------------|
| **Alzheimer's Disease** | |
| Stage 1 | Someone is healthy with no symptoms. |
| Stage 2 | Someone occasionally worries about minor forgetfulness. |
| Stage 3 | Someone frequently misplaces items and struggles with recent memory. |
| Stage 4 | Someone forgets recent conversations and repeats questions. |
| Stage 5 | Someone struggles significantly with language and daily tasks. |
| Stage 6 | Someone regularly becomes confused and has trouble recognizing family. |
| Stage 7 | Someone has severe cognitive decline and is unable to communicate clearly. |
| Stage 8 | Someone is bedridden, minimally responsive, after severe cognitive decline. |
| Stage 9 | Someone just died from severe cognitive decline. |
| **COVID-19** | |
| Stage 1 | Someone is healthy with no symptoms. |
| Stage 2 | Someone has mild fatigue. |
| Stage 3 | Someone has mild fatigue, slight fever, dry cough. |
| Stage 4 | Someone has persistent cough, fever, mild shortness of breath. |
| Stage 5 | Someone has worsening respiratory distress, oxygen saturation 90%. |
| Stage 6 | Someone has been receiving supplemental oxygen and is in the ICU. |
| Stage 7 | Someone was sedated, intubated and mechanically ventilated. |
| Stage 8 | Someone experienced multi-organ failure and declining vital signs. |
| Stage 9 | Someone just died from multi-organ failure and declining vital signs. |
| **COPD** | |
| Stage 1 | Someone is healthy with no symptoms. |
| Stage 2 | Someone experiences shortness of breath with strenuous exercise. |
| Stage 3 | Someone experiences shortness of breath when climbing stairs, mild cough. |
| Stage 4 | Someone is breathless during everyday tasks, chronic cough. |
| Stage 5 | Someone has regular flare-ups, struggles with daily tasks. |
| Stage 6 | Someone requires frequent hospitalizations due to breathing difficulty. |
| Stage 7 | Someone needs supplemental oxygen at home, has severe breathlessness. |
| Stage 8 | Someone experiences chronic respiratory failure, continuous oxygen dependence. |
| Stage 9 | Someone just died from chronic respiratory failure, continuous oxygen dependence. |
| **Parkinson's Disease** | |
| Stage 1 | Someone is healthy with no symptoms. |
| Stage 2 | Someone has mild tremors in one hand. |
| Stage 3 | Someone experiences stiffness and slowness of movement. |
| Stage 4 | Someone has mild difficulty with balance and coordination. |
| Stage 5 | Someone has moderate difficulty with balance and coordination. |
| Stage 6 | Someone has severe tremors and muscle rigidity. |
| Stage 7 | Someone is unable to walk without assistance. |
| Stage 8 | Someone is bedridden with severe motor impairment and requires full-time care. |
| Stage 9 | Someone just died from severe motor impairment and complications. |

# B  ADDITIONAL PERFORMANCE METRICS

Table 6: Per-concept interpretability metrics across models. Values are scores at the last layer (representations right before output).

| Concept | Metric | Llama 70B | Gemma 27B | MedGemma 27B | Qwen 32B | GPT-OSS 120B |
|---|---|---|---|---|---|---|
| Age | $R^2$ (linear) ↑ | **0.98 ± 0.02** | 0.63 ± 0.07 | 0.90 ± 0.06 | 0.59 ± 0.37 | 0.98 ± 0.00 |
| Symptoms | Silhouette ↑ | -0.14 ± 0.05 | -0.13 ± 0.07 | -0.16 ± 0.07 | -0.01 ± 0.04 | **0.01 ± 0.05** |
| Diseases | Silhouette ↑ | -0.03 ± 0.04 | -0.01 ± 0.04 | -0.03 ± 0.04 | **-0.01 ± 0.05** | -0.04 ± 0.03 |
| Disease Progression | CSFS ↓ | 7.50 ± 1.12 | **5.00 ± 1.22** | **5.00 ± 1.22** | 6.25 ± 1.48 | 7.25 ± 2.05 |
| Disease Progression | CSLS ↑ | 4.50 ± 1.66 | **5.00 ± 2.12** | 4.00 ± 1.87 | 4.50 ± 1.66 | **5.00 ± 1.87** |
| Drugs | Silhouette (mech.) ↑ | **0.06 ± 0.08** | -0.03 ± 0.09 | -0.01 ± 0.08 | 0.00 ± 0.06 | 0.03 ± 0.07 |
| Drugs | Silhouette (spec.) ↑ | **0.16 ± 0.13** | -0.03 ± 0.12 | -0.02 ± 0.10 | -0.01 ± 0.07 | 0.04 ± 0.07 |
| Dosages | Patching Effect ↑ | 2.64 ± 1.36 | 0.67 ± 0.82 | 0.98 ± 0.63 | **5.77 ± 9.05** | 2.40 ± 3.64 |

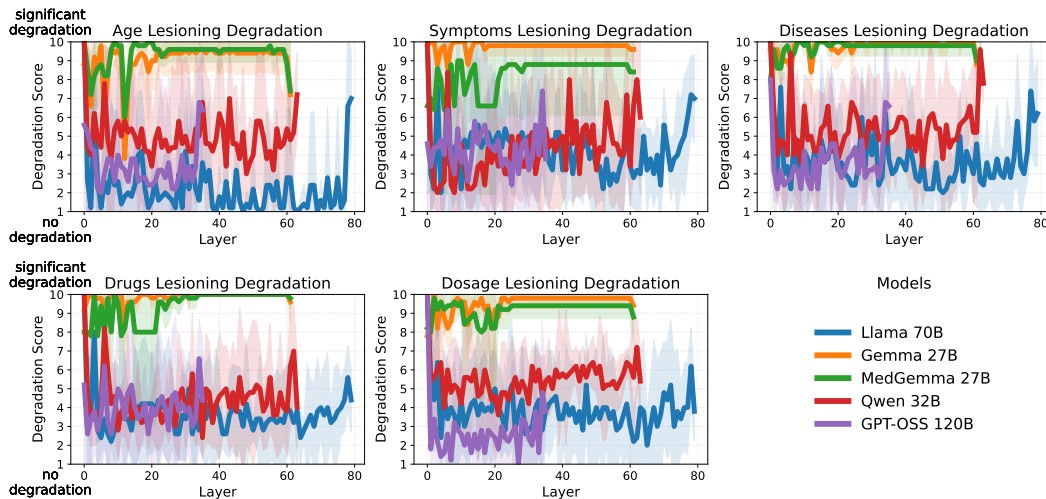

Figure 9: Lesioning metrics per layer.

# C  MAIN LLM MAPS FOR ALL OTHER MODELS

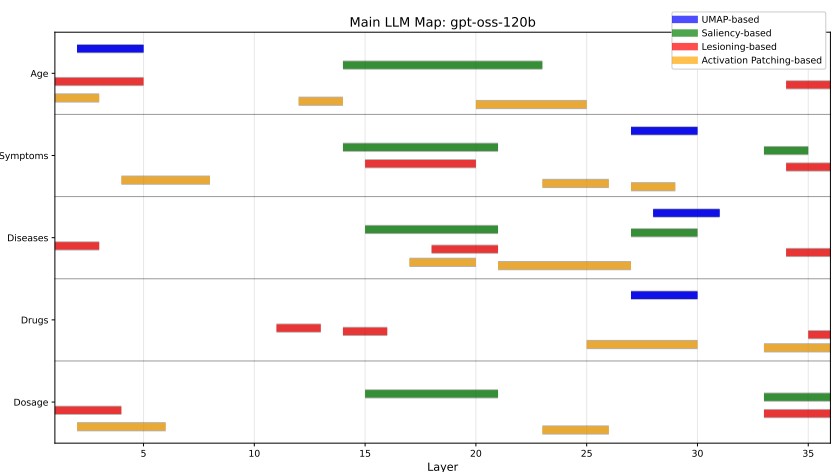

Figure 10: LLM Medical Map for GPT-OSS-120B

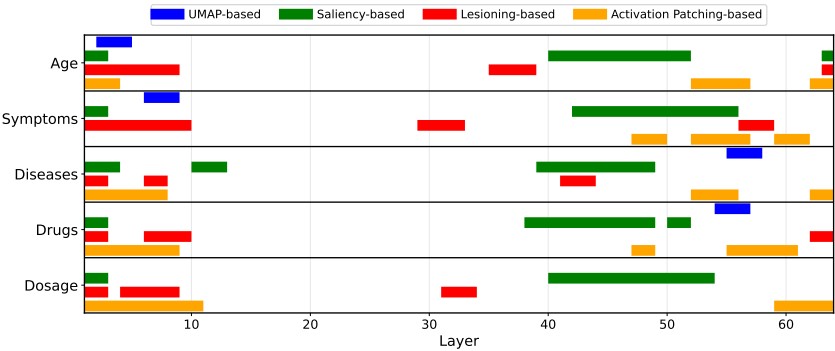

Figure 11: LLM Medical Map for Qwen-32B

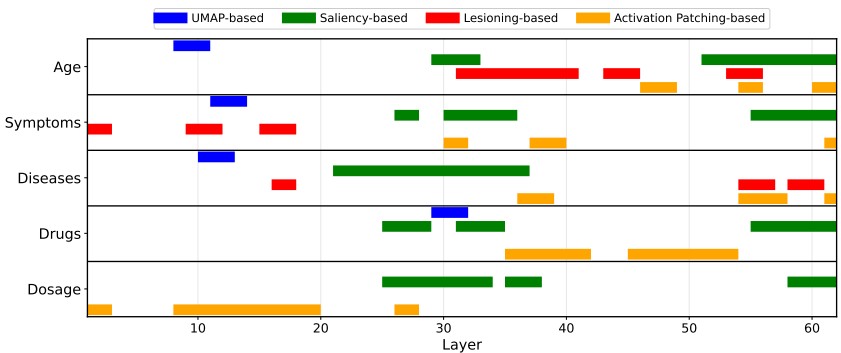

Figure 12: LLM Medical Map for Gemma3-27B

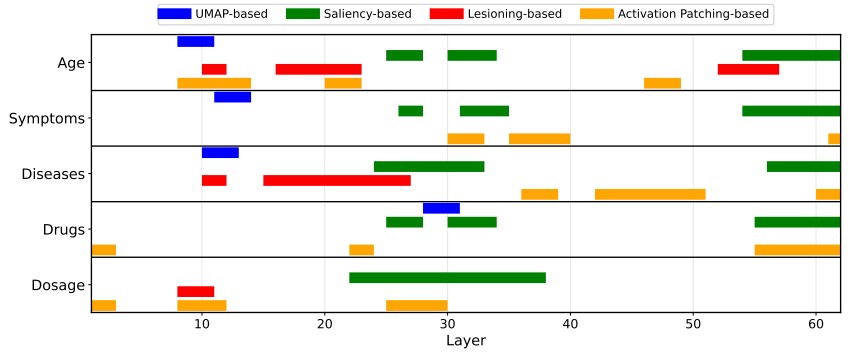

Figure 13: LLM Medical Map for MedGemma-27B

# D FULL PLOTS FOR ALL KEY RESULTS IN MAIN PAPER

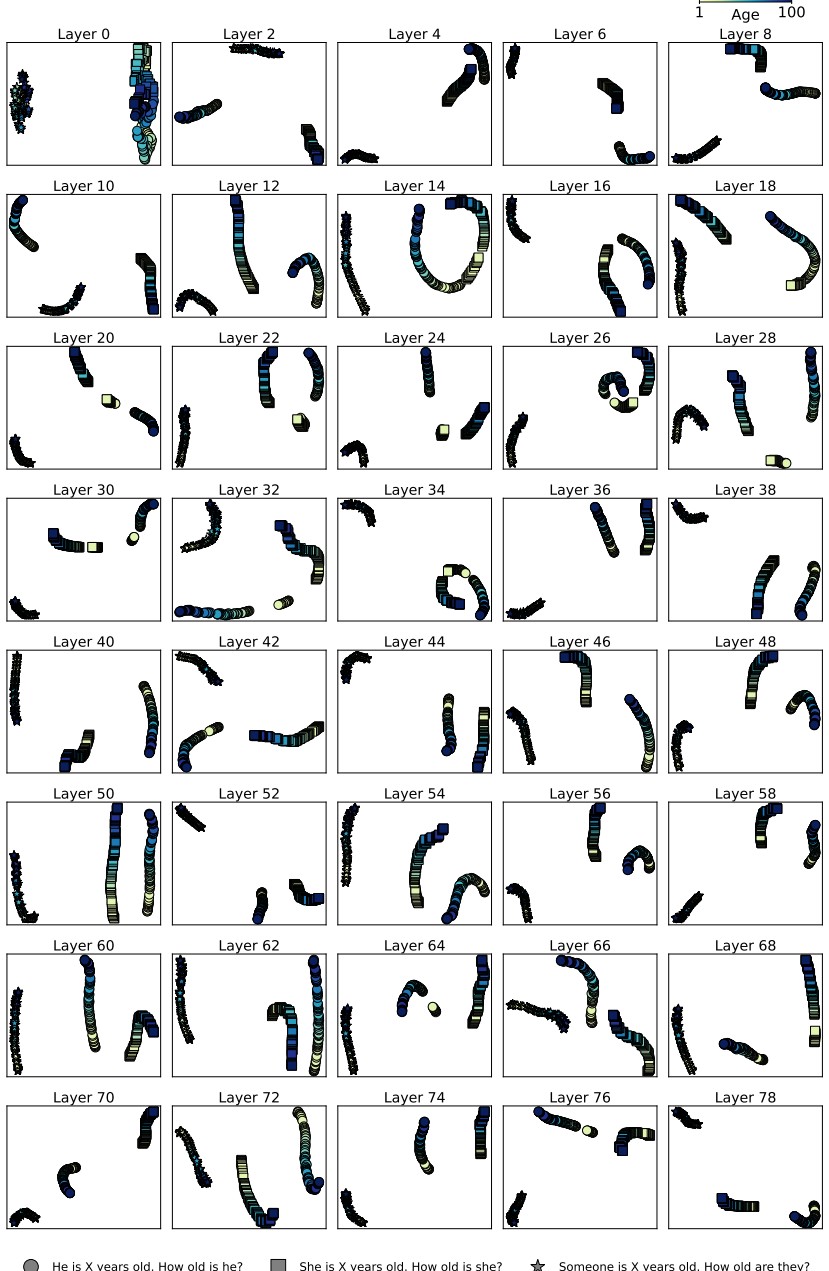

Figure 14: UMAP Analysis for Llama3.3-70B using prompts about subjects with different ages. The age manifold shows non-linearities throughout many intermediate layers (e.g. layer 8). Prompts used are shown at the bottom of the figure.

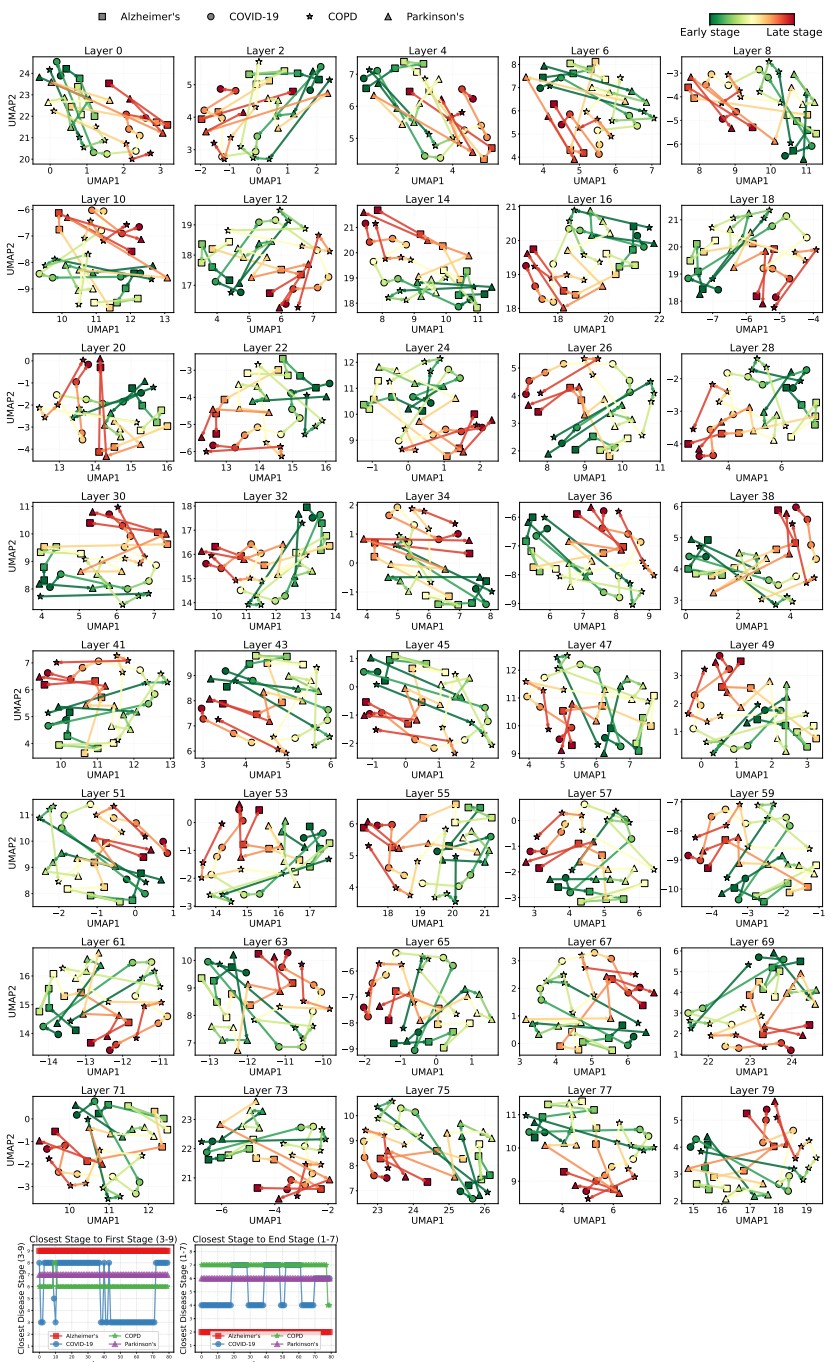

Figure 15: UMAP embeddings for disease progression for Llama3.3-70B .

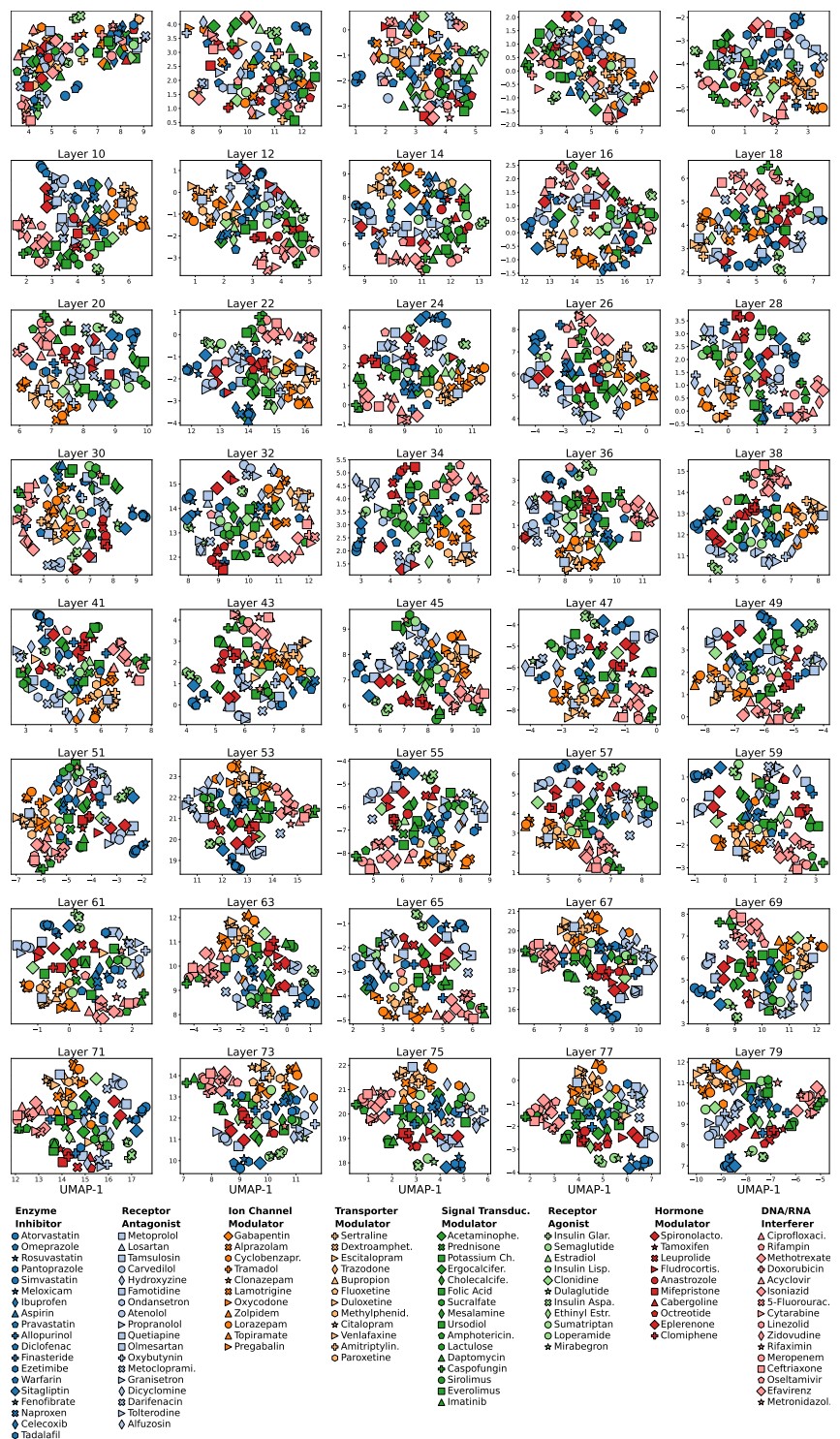

Figure 16: Drug UMAP embeddings by mechanism of action in Llama3.3-70B

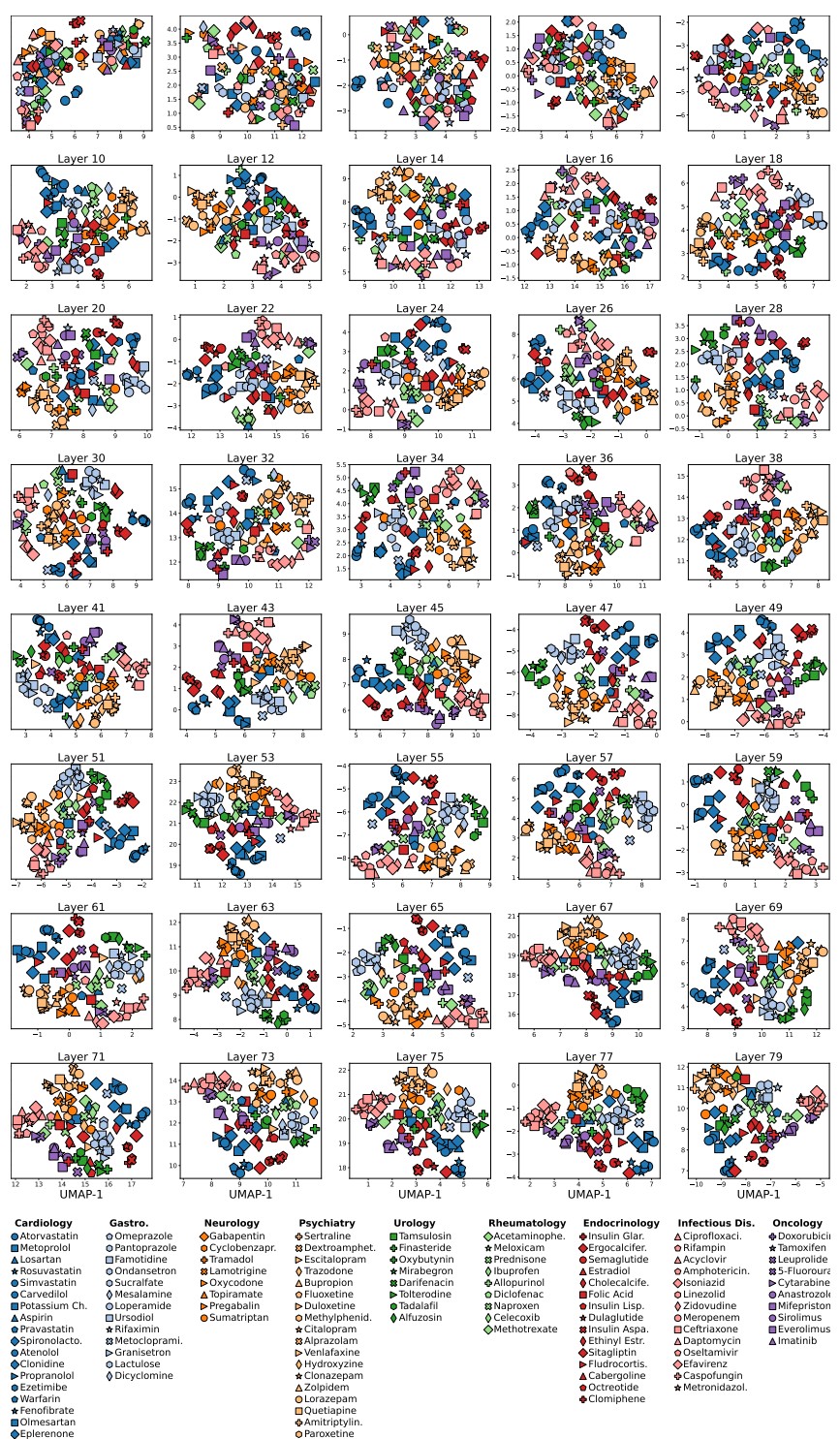

Figure 17: Drug UMAP embeddings by medical specialty in Llama3.3-70B

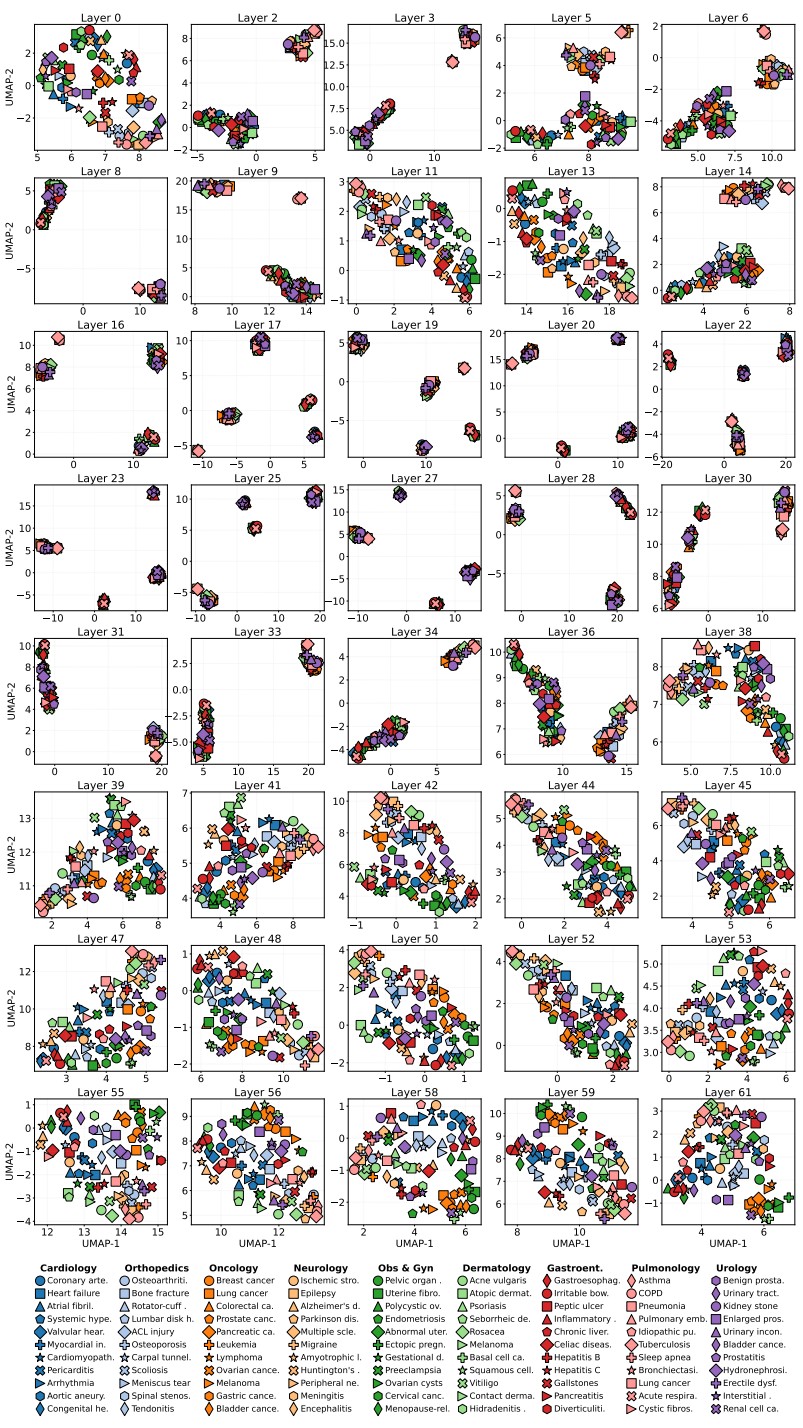

Figure 18: Intermediate layer activations collapse in UMAP space for Gemma.

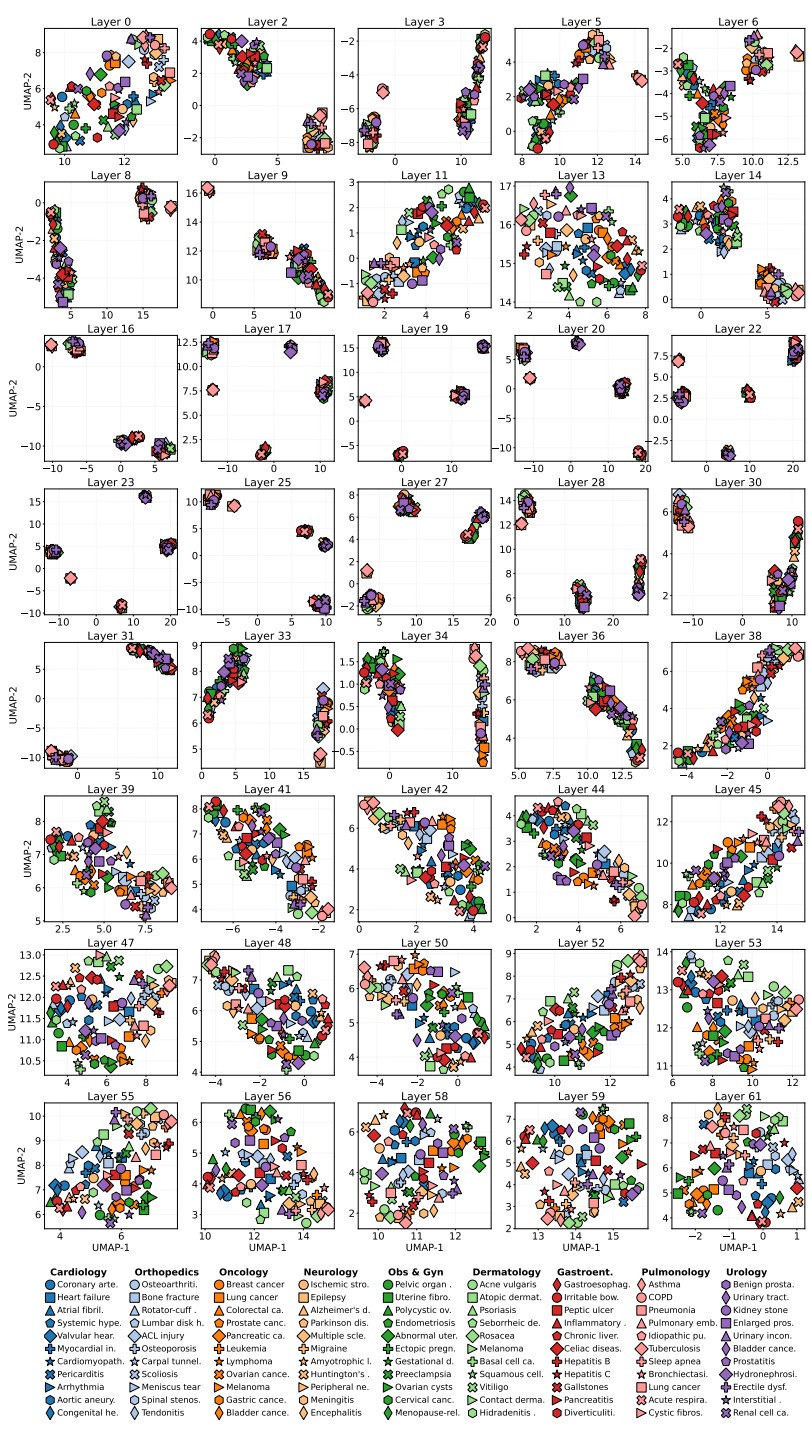

Figure 19: Intermediate layer activations collapse in UMAP space for MedGemma.

# E OTHER UMAP RESULTS ON LLAMA3.3-70B

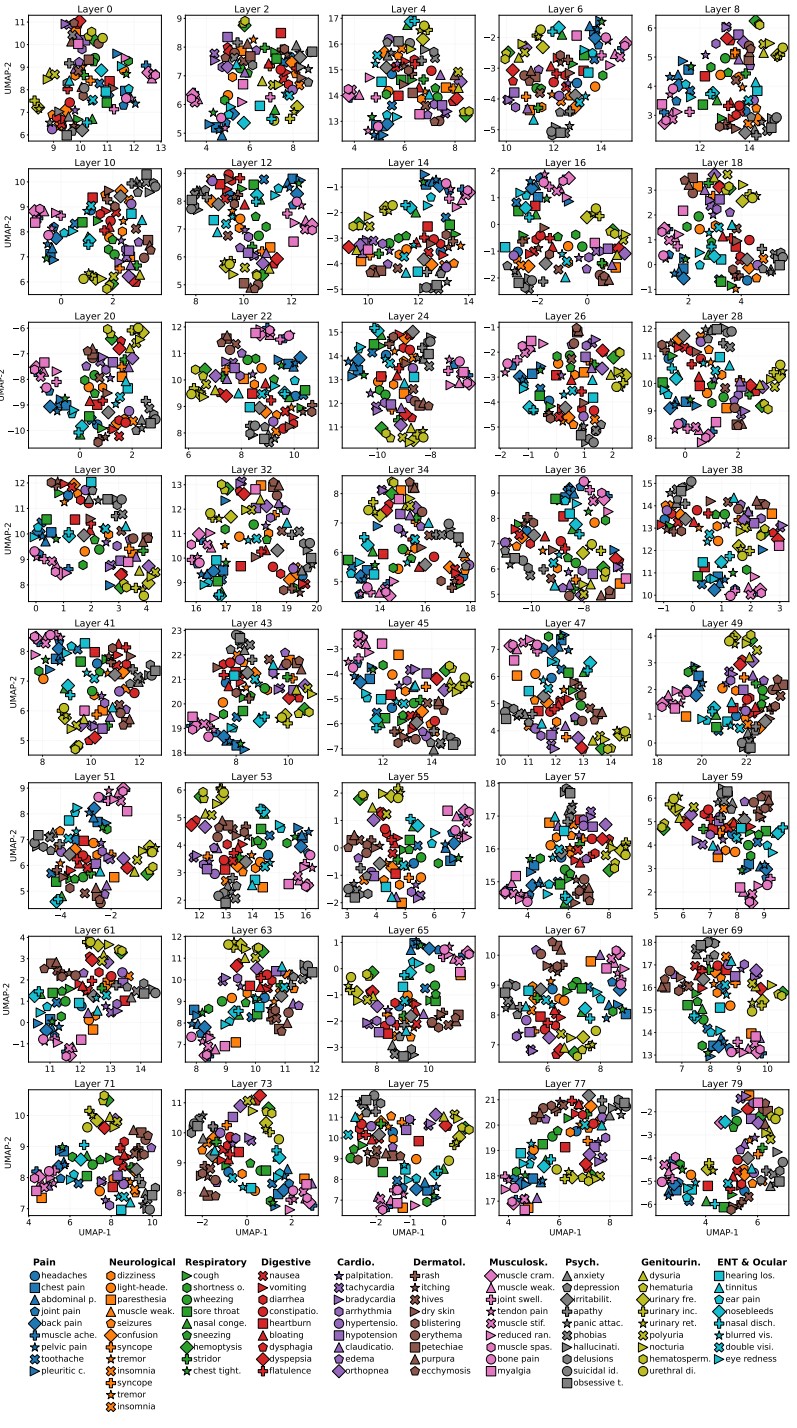

Figure 20: Symptom UMAP embeddings for Llama3.3-70B coloured by symptom groups.

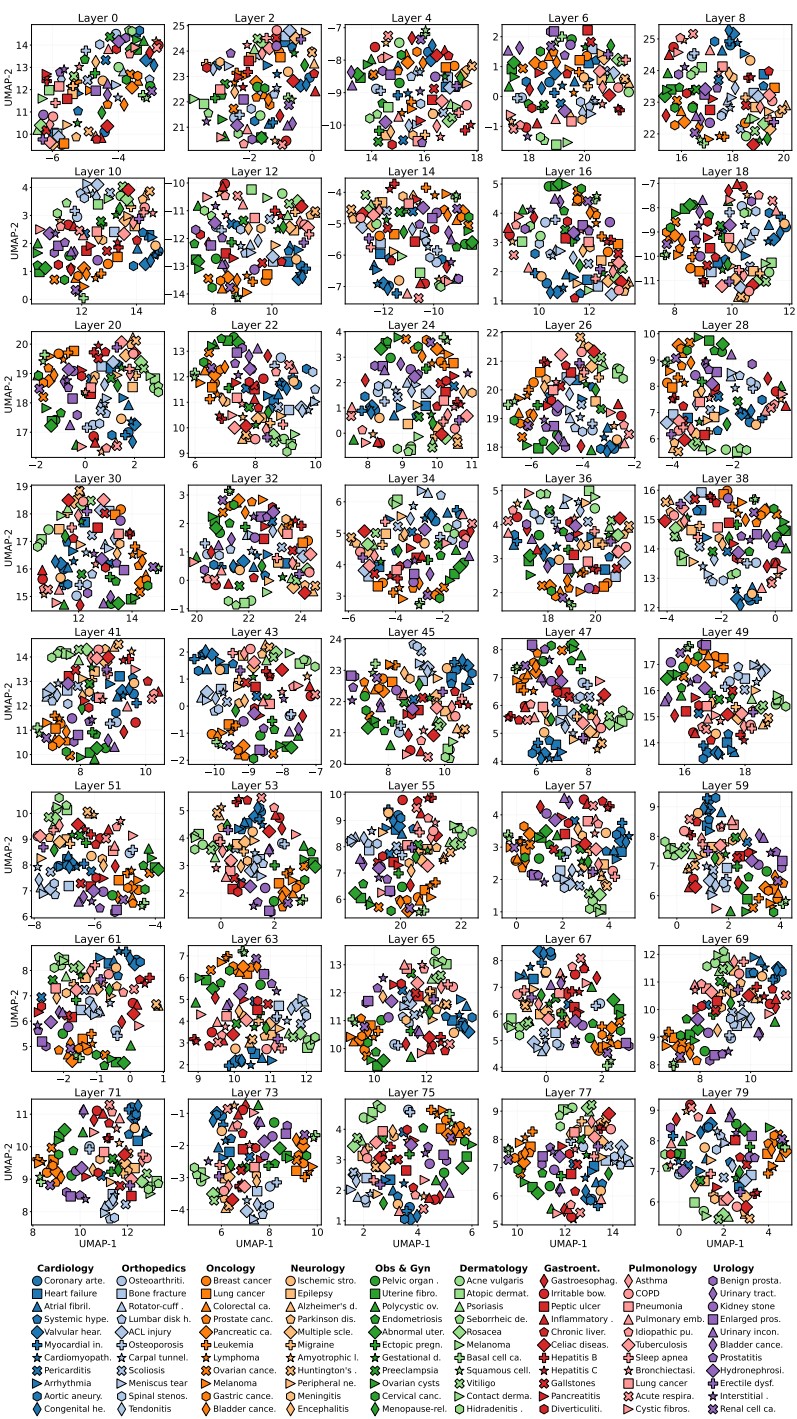

Figure 21: Disease UMAP embeddings for Llama3.3-70B coloured by medical specialties.

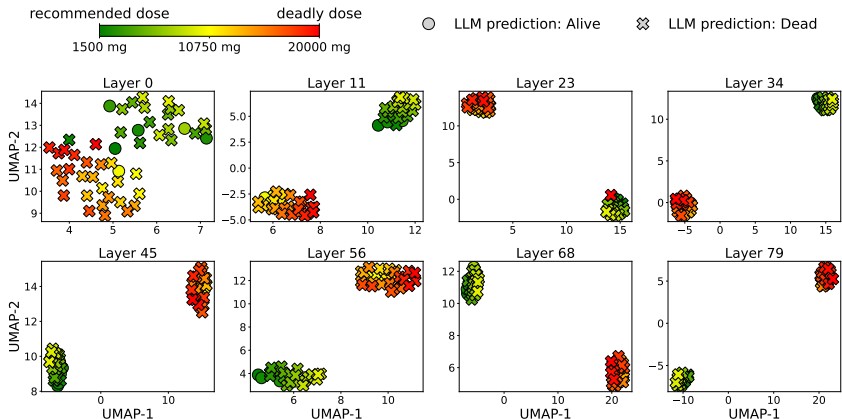

Figure 22: UMAP Analysis for Llama3.3-70B using prompts about different drug dosages for Potassium Chloride. We ran 50 prompts with different drug dosages between the recommended dose (1,500mg) and a deadly dose of 20,000mg. We also analyzed the Llama's response to each dose, and classified it as alive or dead. Two groups are forming early on in the internal representation of the LLM, but these are not aligned with alive/dead groups. We don't understand this behaviour, and further research is needed to look into this.

# F  LLAMA3.3-70B

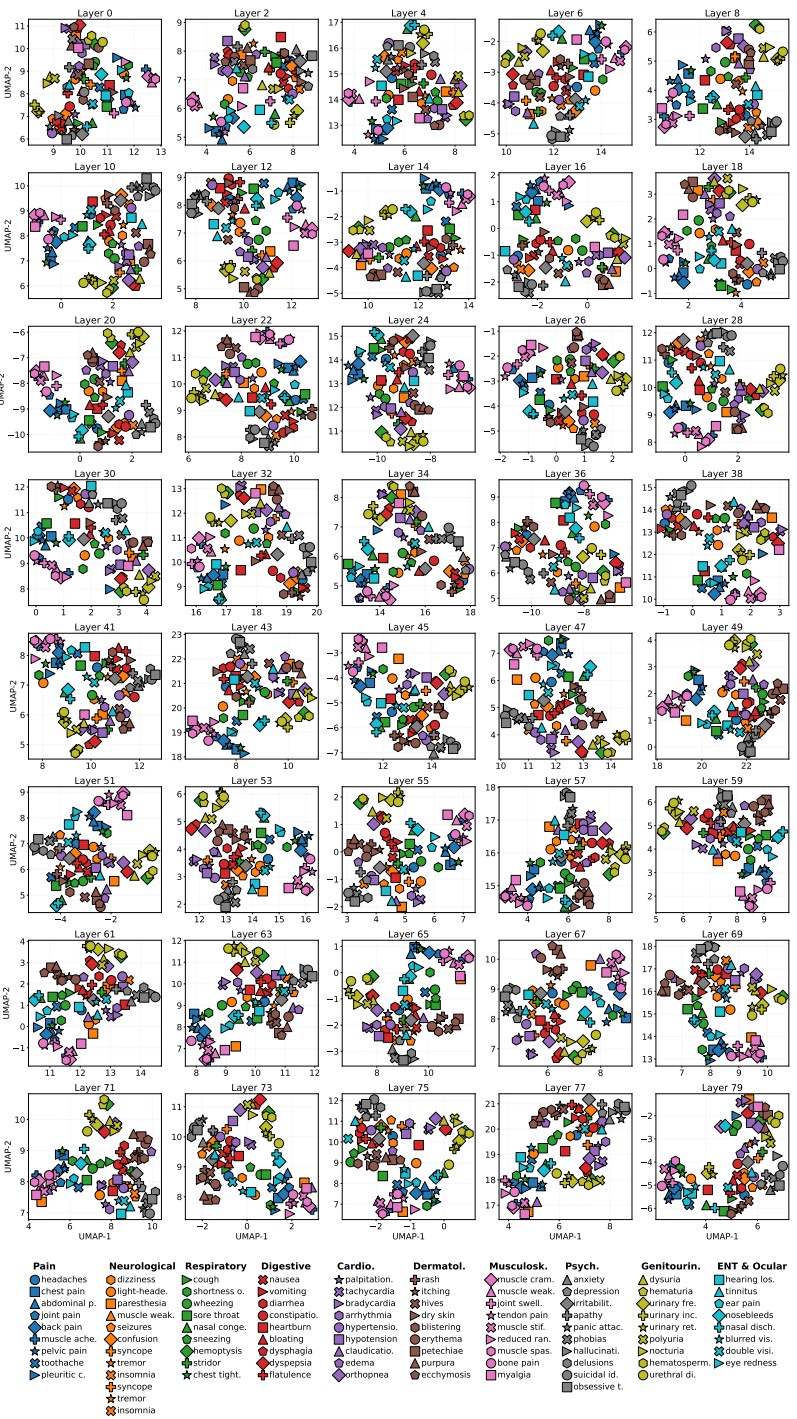

Figure 23: UMAP Analysis for Llama3.3-70B using prompts about different symptoms.

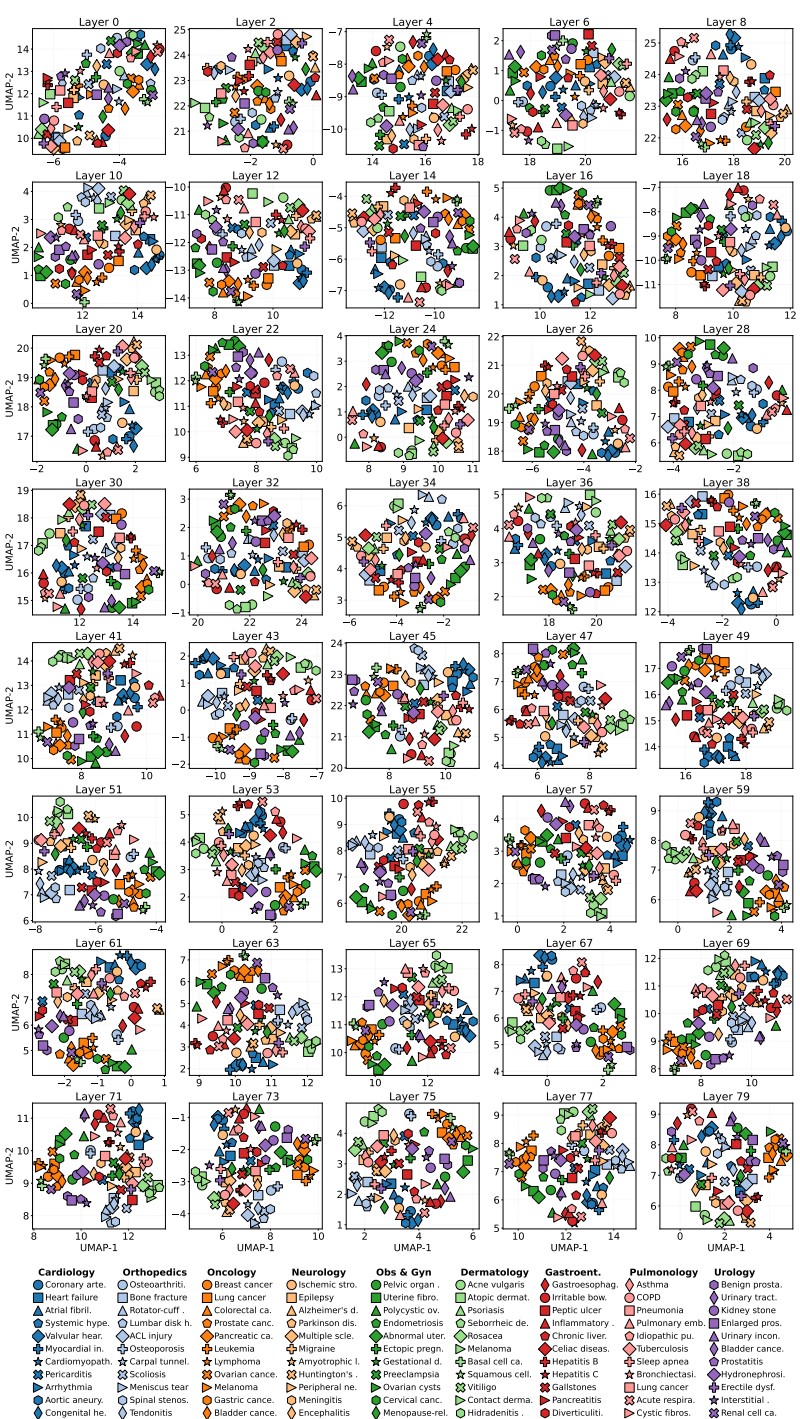

Figure 24: UMAP Analysis for Llama3.3-70B using prompts about different diseases.

# G RESULTS ON OTHER MEDICAL AND LIFE SCIENCE LLMS

Table 7: Per-concept interpretability metrics across models. Values are mean ± std across all layers.

| Concept | Metric | Llama3 OpenBioLLM 70B | PMC LLaMA 13B | Clinical Camel 70B | Palmyra Med 70B | Meditron 70B | HuatuoGPT o1 70B |
|---|---|---|---|---|---|---|---|
| Age | $R^2$ (linear) ↑ | 0.94 ± 0.07 | 0.87 ± 0.12 | 0.92 ± 0.10 | 0.97 ± 0.06 | 0.90 ± 0.12 | **0.98 ± 0.06** |
| Symptoms | Silhouette ↑ | 0.09 ± 0.13 | **0.10 ± 0.06** | 0.06 ± 0.09 | 0.09 ± 0.09 | 0.07 ± 0.10 | 0.08 ± 0.08 |
| Diseases | Silhouette ↑ | -0.01 ± 0.07 | 0.04 ± 0.05 | 0.06 ± 0.06 | **0.06 ± 0.07** | 0.06 ± 0.06 | 0.04 ± 0.06 |
| Disease Progression | CSFS ↓ | **6.74 ± 1.91** | 6.99 ± 1.30 | 6.89 ± 1.30 | 6.82 ± 1.88 | 6.87 ± 1.24 | 7.11 ± 1.64 |
| Disease Progression | CSLS ↑ | 5.26 ± 2.02 | 5.16 ± 1.85 | 5.61 ± 1.57 | 4.98 ± 1.98 | **5.66 ± 1.57** | 5.13 ± 2.01 |
| Drugs | Silhouette (mech.) ↑ | -0.02 ± 0.06 | 0.03 ± 0.05 | 0.04 ± 0.07 | 0.04 ± 0.05 | 0.03 ± 0.07 | **0.05 ± 0.05** |
| Drugs | Silhouette (spec.) ↑ | 0.03 ± 0.10 | 0.05 ± 0.07 | 0.07 ± 0.10 | 0.10 ± 0.10 | 0.07 ± 0.11 | **0.13 ± 0.11** |
| Dosages | Patching Effect ↑ | 3.10 ± 1.15 | -3.43 ± 12.65 | **12.06 ± 25.75** | -0.16 ± 0.67 | 1.69 ± 2.81 | 1.70 ± 0.92 |

Table 8: Per-concept interpretability metrics across models. Values are scores at the last layer (representations right before output).

| Concept | Metric | Llama3 OpenBioLLM 70B | PMC LLaMA 13B | Clinical Camel 70B | Palmyra Med 70B | Meditron 70B | HuatuoGPT o1 70B |
|---|---|---|---|---|---|---|---|
| Age | $R^2$ (linear) ↑ | 0.98 ± 0.01 | 0.91 ± 0.07 | 0.93 ± 0.05 | 0.98 ± 0.02 | 0.86 ± 0.12 | **0.98 ± 0.01** |
| Symptoms | Silhouette ↑ | -0.07 ± 0.05 | **0.02 ± 0.04** | -0.12 ± 0.03 | -0.10 ± 0.04 | -0.02 ± 0.04 | -0.11 ± 0.04 |
| Diseases | Silhouette ↑ | -0.07 ± 0.03 | **0.00 ± 0.04** | -0.06 ± 0.04 | -0.03 ± 0.04 | -0.01 ± 0.04 | -0.06 ± 0.04 |
| Disease Progression | CSFS ↓ | 7.50 ± 1.12 | 7.25 ± 1.48 | 7.75 ± 2.17 | **7.00 ± 1.87** | 7.25 ± 1.30 | 7.50 ± 1.12 |
| Disease Progression | CSLS ↑ | 5.25 ± 1.92 | 6.25 ± 0.43 | **6.50 ± 0.50** | 4.50 ± 1.66 | **6.50 ± 0.50** | 4.50 ± 1.66 |
| Drugs | Silhouette (mech.) ↑ | 0.02 ± 0.06 | 0.05 ± 0.05 | 0.07 ± 0.07 | 0.07 ± 0.05 | 0.08 ± 0.07 | **0.09 ± 0.05** |
| Drugs | Silhouette (spec.) ↑ | 0.07 ± 0.10 | 0.08 ± 0.07 | 0.09 ± 0.10 | 0.15 ± 0.10 | 0.11 ± 0.11 | **0.17 ± 0.11** |
| Dosages | Patching Effect ↑ | 3.22 ± 1.20 | 2.75 ± 3.97 | **11.89 ± 24.77** | 0.26 ± 0.56 | 1.70 ± 2.74 | 1.78 ± 0.94 |

