# OpenReview forum: "Medical Interpretability and Knowledge Maps of Large Language Models"
_ICLR.cc/2026/Conference — ICLR 2026 Poster_

### Official Review · Reviewer_bgnx · 2025-10-28

**Soundness:** 3
**Presentation:** 2
**Contribution:** 3
**Rating:** 4
**Confidence:** 2

**Summary:**

This paper has focused on the medical-domain interpretability in large language models. The authors have found that existing works have only explored limited medical knowledge areas with a single explainability technique. To fill the gap in this field, this paper has explored four interpretability methods on five various LLMs. Through extensive experiments, the authors have found several conclusions.

**Strengths:**

+ S1. This paper is well-organized and well-written, making it easy to follow.
+ S2. Extensive experiments have been conducted.
+ S3. The code is released, making it easy to reproduce.

**Weaknesses:**

- W1. The motivation for the specific exploration in medical areas is insufficient. It is still unclear why the interpretability techniques cannot be well adopted in the medical area.
- W2. The authors have claimed that they focused on the medical domain, but only one medical-specific LLM is experimented with in this paper, i.e., MedGemma-27B. In my view, more related LLMs, such as Huatuo-GPT, should be considered in this paper, instead of general-purpose LLMs.
- W3. Though the authors have argued that previous works only consider one of the medical knowledge areas. However, this paper also only considers them independently, while ignoring the relationships between them. Thus, it seems that this paper has only conducted more experiments but has not addressed this issue basically.
- W4. Why are the four interpretability methods, i.e., UMAP, gradient-based saliency, layer lesioning, and activation patching, selected in this paper? What's the selection criterion?

**Questions:**

All my questions have been included in the weakness section.

---

> ### Author Response · Authors · 2025-11-20
> **Thank you for the review**
>
> We thank the reviewer for the feedback and offer responses below.
>
> ===================================================================
>
> [W1] Unclear why interpretability techniques cannot be well adopted in the medical area:
>
> We never claimed this in our paper. We are actually using standard interpretability techniques, and we found that they work relatively well on LLMs.
>
> ===================================================================
>
> [W2] Only one medical-specific LLM is experimented with in this paper
>
> This is an interesting point. We chose those LLMs due to being widely used and forming the backbone for other domain-specific LLMs. However, we have since ran 6 additional medical/biology-specific LLMs: HuatuoGPT-70B, Llama3-OpenBioLLM-70B, PMC_LLaMA_13B, ClinicalCamel-70B, Palmyra-Med-70B and Meditron-70B. We obtained the following metrics for these models.
>
> | Concept             | Metric                     | Llama3 OpenBioLLM 70B | PMC LLaMA 13B | Clinical Camel 70B | Palmyra Med 70B | Meditron 70B | HuatuoGPT o1 70B |
> |---------------------|-----------------------------|------------------------|----------------|----------------------|-------------------|----------------|-------------------|
> | Age                 | R² (linear) ↑               | 0.94 ± 0.07            | 0.87 ± 0.12     | 0.92 ± 0.10          | 0.97 ± 0.06       | 0.90 ± 0.12    | **0.98 ± 0.06**   |
> | Symptoms            | Silhouette ↑                | 0.09 ± 0.13            | **0.10 ± 0.06** | 0.06 ± 0.09          | 0.09 ± 0.09       | 0.07 ± 0.10    | 0.08 ± 0.08       |
> | Diseases            | Silhouette ↑                | -0.01 ± 0.07           | 0.04 ± 0.05     | 0.06 ± 0.06          | **0.06 ± 0.07**   | 0.06 ± 0.06    | 0.04 ± 0.06       |
> | Disease Progression | CSFS ↓                      | **6.74 ± 1.91**        | 6.99 ± 1.30     | 6.89 ± 1.30          | 6.82 ± 1.88       | 6.87 ± 1.24    | 7.11 ± 1.64       |
> | Disease Progression | CSLS ↑                      | 5.26 ± 2.02            | 5.16 ± 1.85     | 5.61 ± 1.57          | 4.98 ± 1.98       | **5.66 ± 1.57**| 5.13 ± 2.01       |
> | Drugs               | Silhouette (mech.) ↑        | -0.02 ± 0.06           | 0.03 ± 0.05     | 0.04 ± 0.07          | 0.04 ± 0.05       | 0.03 ± 0.07    | **0.05 ± 0.05**   |
> | Drugs               | Silhouette (spec.) ↑        | 0.03 ± 0.10            | 0.05 ± 0.07     | 0.07 ± 0.10          | 0.10 ± 0.10       | 0.07 ± 0.11    | **0.13 ± 0.11**   |
> | Dosages             | Patching Effect ↑           | 3.10 ± 1.15            | -3.43 ± 12.65   | **12.06 ± 25.75**    | -0.16 ± 0.67      | 1.69 ± 2.81    | 1.70 ± 0.92       |
>
>
> We also computed two additional tables with metrics at best layer, and at final layer, across all these models. The results are now in Appendix G in the updated paper pdf. We can conclude the following:
> - Across all layers, HuatuoGPT has best age linearity and drug clustering
> - disease progression shows circularity for these models as well. Across all 6 models, CSFS (closest stage to first stage) are > 6.7, far from an ideal value of 2 or 3.
> - activation patching succeeded at some layer for all models other than Palmyra (success = patching effect > 0.5)
>
>
> ======================================================================================
>
> [W4] Why are there four interpretability methods?
>
> This is because each method has its own biases and makes its own assumptions. UMAP assumes intermediate activations lie on a lower-dimensional manifold. Saliency methods perform a first-order Taylor approximation of the loss function around the input datapoint (here input LLM prompt). Activation Patching assumes that only localized causal interventions can change certain outcomes and also assumes a linear interpolation between the clean and corrupted prompt to compute the effect. Layer lesioning is non-specific to medical concepts; for example, taking out layer 3 can degrade performance not just on disease processing, but also on other tasks such as reasoning or knowledge recall. However, when multiple such methods are used and reach overlapping conclusions, the scientific results become significantly stronger. This “convergence of evidence” is called consilience (https://en.wikipedia.org/wiki/Consilience), and is a cornerstone of our study. As opposed to previous interpretability works which run a single method, we run four entirely different methods, and the convergence of evidence from all four methods shows with much higher certainty the layers where most medical knowledge is being processed.
>
> We hope these new experiments and additional clarifications give the reviewer significantly more confidence in our study.

---

> > ### Comment · Reviewer_bgnx · 2025-11-27
> >
> > My concerns have been addressed, so I'd like to raise the score accordingly.

---

### Official Review · Reviewer_W4B3 · 2025-11-01

**Soundness:** 3
**Presentation:** 3
**Contribution:** 3
**Rating:** 6
**Confidence:** 3

**Summary:**

This paper presents an interpretability analysis of five popular LLMs, focusing on the medical knowledge encoded within their layers. The authors employ four interpretability methods: gradient-based saliency, UMAP projections, activation patching, and layer lesioning. They examine how medical knowledge is localized across patient demographics, diseases, and drug treatments.

**Strengths:**

- The paper explores an interesting and typically underrepresented domain within interpretability research.
- It employs multiple complementary approaches, offering a broad and consolidated set of metrics and analysis.

**Weaknesses:**

**[W1]** While the paper presents a substantial number of results and metrics, it falls short in translating these insights into actionable recommendations. Even in the discussion section, the authors primarily offer intuition behind their findings but acknowledge that further analysis is required. To enhance the practical relevance of this work, authors can provide a clearer argumentation or implementations regarding how their analysis could inform model development.

**[W2]** The paper does not specify key statistics or details about the data used to probe the models and conduct the mechanistic analysis (apart from the prompt templates listed in the Appendix). Given that interpretability results can be highly data-dependent, this omission makes it difficult to assess the generalizability of the findings.

**Questions:**

Please refer to the weaknesses.

---

> ### Author Response · Authors · 2025-11-19
> **Thank you for the feedback**
>
> We really appreciate and thank the reviewer for the feedback and for carefully reading our manuscript. We updated the paper pdf based on the reviewer feedback, which can be accessed here on OpenReview for the reviewer to see.
>
> ===== [W1] Paper fails to translate results into actionable recommendations ========
>
> We thank the reviewer for making this point. Due to limited space, we mentioned only in the abstract about potential for fine-tuning, un-learning or de-biasing. Basically, all of these can be done significantly more efficiently by targeting the precise layers that our analysis uncovered. We give below three examples of actionable recommendations, which we have included in section 4.1 of the paper, which will come in the extra page in the camera-ready version:
>
> Removing age discontinuities and enforcing linear manifold for ages: If one wants to de-bias the age in order to remove the discontinuity at 18 years or to make the age manifold entirely linear, one can fine-tune only those exact layers where age is processed and use an age regularizer in the loss function that maintains linearity, such as $\mathcal{L_\text{age}} = \lambda * (1 - \text{R}(\text{ages}, \text{UMAP}(\text{prompt}(\text{ages}))))$, where $R$ is the linear regression coefficient which will be 1 if the manifold of age embeddings is entirely linear.
>
> Preventing collapse of Gemma/MedGemma: To prevent the collapse of Gemma/MedGemma, the Google team can add during training a Uniformity loss on a hypersphere (Wang & Isola, 2020, https://arxiv.org/abs/2005.10242) as follows: $\mathcal{L_{unif}} = log\ \mathbb{E}_{i \neq j}\ ​\text{exp}(−\alpha || z_i​−z_j​ || ^ 2)$. A large $\alpha$ will incur a large penalty for small distances between embeddings $z_i$ and $z_j$, thus preventing the collapse into 3-4 blobs as seen in the UMAP Figures.
>
> Enforcing monotonicity in disease progression: During training, AI labs can add a regularizer that enforces monotonic disease progression, such as:
>
> $\mathcal{R_{monotonic}} = \lambda \sum_{s=1}^{S-1} \left[ \max\left(0,\; d(h_{s}, h_{1}) - d(h_{s+1}, h_{1}) \right) \right]^2$
>
> which enforces the hidden-state activations $h_s$ at disease stage $s$ to move further away from $h_1$, the activation for stage 1 (healthy). These can be made even more complex by anchoring not just at $h_1$, but at other intermediate stages as well.
>
> ====== [W2] Add key statistics about the data ==============================
>
> We are not entirely clear what kind of statistics would actually be useful to report. If the reviewer would suggest some such statistics along with a short explanation for their usefulness, we would be happy to add them to the paper.
>
> We hope these answers clarify the results and impact of our paper, and can give more confidence to the reviewer.

---

### Official Review · Reviewer_Denk · 2025-11-02

[review text omitted: it was posted to a different submission]

---

> ### Author Response · Authors · 2025-11-19
> **Review is for the wrong paper**
>
> We thank the reviewer, but we would like to point out that this review is not for our paper. We never proposed CPA (Causal Prototype Alignment), never performed causal alignment, nor counterfactual consistency tests. We also didn’t run any experiments on MIMIC-III. We believe this review was meant for a different paper. If the reviewer has a review for our paper, we would be happy to read it and address it.

---

> > ### Comment · Reviewer_Denk · 2025-11-23
> >
> > Yes, my apologies — I had the wrong tab open and accidentally wrote the review for a different paper. I’ve corrected it now and updated the review accordingly.

---

> ### Author Response · Authors · 2025-11-25
>
> Thank you very much for updating the review. We list below our answers to the review. We also updated the pdf to include, amongst others, circularity quantitative analyses without UMAP altogether, and quantitative metrics on 6 new medical LLMs (appendix G).
>
> ========================================================
>
> [W1] Validity concerns due to prompts being manually constructed
>
> We agree with this concern. To ensure medical validity, a clinical neuroscientist on our team reviewed the prompt templates, disease-stage definitions, and drug-category labels. This was part of our workflow from the beginning and is noted in the last sentence of the introduction. These expert-validated prompts and labels ensured medically-correct and relevant concepts are being assessed.
>
> ========================================================
>
> [W2, Q1, Q3]. Limitations due to UMAP, which is known to distort topology; can the conclusions such as “circular disease progression” be  supported with additional metrics?
>
> We thank the reviewer for the insightful suggestion. In response, we recomputed the circularity analysis **entirely without UMAP**. The updated results are presented in Fig. 4 (bottom-center and bottom-right) in the updated pdf, using L2 distances directly in the original activation space. The circularity patterns persist and are, in fact, less noisy than in the UMAP-projected space. We retained UMAP only for visualization purposes, but all quantitative circularity claims now rely solely on the high-dimensional geometry.
>
>
> ============================================================
>
> [Q2] How dependent are results on the exact prompt templates?
>
> A systematic prompt-sensitivity and variability study would require running many additional prompts at scale. This entails substantial computational cost, which is beyond the feasible scope of this submission. Our current experiments already cost us close to $8,000 in GPU-hours on a commercial cloud. To remain within reasonable resource and time constraints for an ICLR submission, we prioritized analyses most directly aligned with the paper’s core claims. We agree that a comprehensive variability analysis is valuable future work.
>
> ================================================================
>
> [W3] Ablations: prompt sensitivity, alternative distances, non-medical tasks, generic behavior
>
> Performing the full combinatorial set of ablations requested—covering prompt variability, alternative metrics, non-medical variability such as different gendered pronouns — would require a substantial increase in compute resources and evaluation time. Within the scope of this submission, we opted to prioritize depth on core medical interpretability questions over breadth across all ablations.
>
> To partially address this concern, we have since evaluated six additional medically oriented LLMs through our pipeline. We added three new tables in Appendix G summarizing the interpretability metrics across all six new models. The consistency of several observed patterns across these models increases confidence that the findings are not artifacts present in a single LLM.
>
> ==========================================================
>
> Extras added now at review stage
>
> In section 4.1 of the updated manuscript pdf (available here on openreview), we now provided actionable recommendations for LLM training based on our findings. This includes how to remove discontinuities and enforcing linearity in the age manifold, how to prevent the collapse in Gemma/MedGemma, and how to enforce monotonicity in disease progression.
>
> ==========================================================
>
> We would kindly appreciate it if the reviewer updates the summary section of the main comment as well, as it's still about the earlier wrong paper.

---

> > ### Comment · Reviewer_Denk · 2025-11-25
> > **Summary updated**
> >
> > Thank you for the detailed reply. I'll check it later but revise the summary first.

---

> > > ### Comment · Area_Chair_LgHh · 2025-11-30
> > >
> > > Dear Authors,
> > >
> > > Do you have any record on the updated review?

---

> > > > ### Author Response · Authors · 2025-12-04
> > > >
> > > > We unfortunately do not have a record of it, but the reviewer likely has it and might be able to re-post it.

---

> > > > > ### Comment · Area_Chair_LgHh · 2025-12-04
> > > > >
> > > > > Reviewer is not allowed to do that

---

### Official Review · Reviewer_QKiR · 2025-11-06

**Soundness:** 3
**Presentation:** 3
**Contribution:** 3
**Rating:** 6
**Confidence:** 3

**Summary:**

This paper presents a systematic study of medical interpretability in large language models (LLMs), introducing “medical knowledge maps” across five open-source models. It combines four complementary interpretability methods—UMAP projections, gradient-based saliency, layer lesioning, and activation patching—to locate where medical knowledge (age, symptoms, diseases, drugs, dosages) is stored within model layers. The work is ambitious and methodologically comprehensive, providing interesting insights into layer-wise organization and representational phenomena such as non-linear age encoding and circular disease progression.

**Strengths:**

* The study systematically applies four complementary interpretability methods, yielding a triangulated view of where medical knowledge resides in LLM layers.
* The work addresses a relatively underexplored area by focusing interpretability specifically on medical-domain knowledge and tasks.
* The empirical scope spans five models and multiple medical subdomains, improving the breadth and potential generalizability of the findings.
* The paper presents clear visualizations and “LLM maps,” making the layer-wise organization of knowledge intuitive to interpret.
* The insights have practical implications by potentially guiding fine-tuning, unlearning, and bias-mitigation strategies for medical LLMs.

**Weaknesses:**

* The evaluation lacks strong external validation and limited use of ground truth, leaving some claims insufficiently anchored beyond internal metrics.
* The interpretation of non-linear and circular manifolds remains conceptually ambiguous, risking over-interpretation of representational geometry.
* The statistical analysis does not thoroughly quantify robustness across seeds, prompt perturbations, or hyperparameter choices, which weakens rigor.
* The connection to broader interpretability theory and causal abstraction frameworks is not fully developed, reducing theoretical grounding.
* The reproducibility story in the main text omits detailed settings and implementation specifics (e.g., UMAP and patching parameters), which may hinder independent replication.

**Questions:**

* How stable are the LLM maps across different prompt templates or random seeds?
* Did you consider potential confounding due to tokenization or vocabulary frequency in medical terms?
* Could the observed circular disease progression simply arise from UMAP distortions rather than true representational topology?
* Have you tested whether the identified “knowledge layers” align with known reasoning behaviors (e.g., zero-shot diagnosis tasks)?

---

> ### Author Response · Authors · 2025-11-19
> **Thank you for the feedback**
>
> We really appreciate and thank the reviewer for the feedback and for carefully reading our manuscript. We updated the paper pdf based on the reviewer feedback, which can be accessed here on OpenReview for the reviewer to see.
>
> ====================================================================
>
> (W1) Lack of strong external validation and use of limited ground truth
>
> We discussed this in the limitations section at the end of the paper. While we already performed significant quantitative evaluation (we provided three tables of metrics and a figure with 12 sub-plots), and we used four entirely different methods which through consilience give much stronger evidence than if only a single method is used. We don't see exactly how one would run an external validation in our case and what that would bring. If the reviewer has any specific suggestions, we'd be happy to discuss them. In addition, using a ground truth setting would require re-training an LLM such as Llama-70B where we choose to fine-tune certain layers (say layers 4-10) for processing e.g. diseases, and repeat this for other medical tasks by choosing other layers to be fine-tuned. Given that training Llama-70B took 7.0 million H100 hours, we do not have that computational ability.  If the reviewer would kindly suggest any additional way to validate the results that we could realistically implement, we would be more than happy to do it.
>
> ====================================================================
>
> (W3, Q1) Stability of LLM maps across different prompts, random seeds, hyper-parameters
>
> We did not attempt this additional exploration due to the sheer nature of the task. The experiments we ran with 5 different analyses types (age, symptoms, disease, drugs, drug dosages) x 5 LLMs already used significant compute resources (currently close to $8,000 on GCP). We simply do not have the resources to significantly explore those variations.
>
> ===================================================================
>
>  (W4) Connection to border interpretability theory and causal abstraction frameworks
>
> Our current study was an empirical study, with the key aim of finding and mapping where medical knowledge resides inside LLMs. We agree it is highly important to map our findings to more causal mechanisms, perhaps uncover circuits within the LLMs for processing such medical tasks, but those are highly significant works in themselves which we can do in a future study.
>
> ====================================================================
>
> (W5) Omitted settings related to UMAP and patching parameters
>
> We agree, and we have now included in the paper the full parameters of the dimensionality reduction of UMAP (n_neighbours =15, min_dist=0.1, cosine distance metric). For activation patching, we closely followed Zhang and Nanda, 2024 where the only hyper-parameter we are aware of is the threshold for the patching to be considered successful (which we set to 0.5 and clearly mentioned it in the main text). We additionally provided the code and will publish the GitHub repo, which will allow the work to be easily reproducible.
>
> ====================================================================
>
> (Q2) Confounding due to tokenization or vocabulary frequency in medical terms
>
> Again, due to resource limitations, we did not explore these additional dimensions of variability. A big AI lab with significantly more resources would need to follow up on our work.
>
> ====================================================================
>
> (Q3) Can circular disease progression be caused by UMAP distortions?
>
> We thank the reviewer for this observation. We re-computed the scores for circularity entirely without UMAP, using the activations in the original space, and the results still hold, and are actually even more clear to interpret and less noisy. We updated the circularity plots in Fig. 4 (bottom-center and bottom-right), which now show more clearly that all diseases show circularity, with Alzheimer's showing the least amount of such circularity. We get similar results with the 6 new biomedical LLMs we just ran in response to another reviewer (see Appendix G, disease progression metrics). We note this is highly significant due to the known monotonic nature of these diseases, suggesting that the LLM intermediate representations are biased.
>
> Overall, we thank the reviewer for the comments, and if there are additional clarifications or experiments we can do to give the reviewer more confidence, we'd be happy to run them.

---

### Meta-Review · Area_Chair_LgHh · 2025-12-20

**Summary:**

This paper presents an interpretability study of medical-domain knowledge in LLMs. The authors analyse five open-source LLMs(Llama3.3-70B, Gemma/MedGemma-27B, Qwen-32B, GPT-OSS-120B) using four complementary interpretability techniques, UMAP projections, gradient-based saliency, layer lesioning, and activation patching, to find where medical knowledge maps. They localise age, symptoms, diseases, drug categories, dosages, and disease progression are encoded across model layers.

Three reviewers recommend acceptance in initial review. It is noted that Denk at first upload the review of different paper. The final correct version is now attached in the comment. Reviewer bgnx who initially gave 4 also raised the score after reading the rebuttal.

Reviewers consistently highlight the value of triangulating multiple interpretability methods, the inclusion of multiple models and medical subdomains, and the quality of the visualisations, which make layer-wise representational patterns intuitive. Several empirical observations are considered interesting and potentially impactful, with relevance to fine-tuning, debiasing, unlearning, and medical AI safety.

While reviewers raise concerns regarding rigor, grounding, and interpretation, they are effectively resolved in rebuttal. Therefore, this paper is recommend for acceptance.

**Reviewer Concerns:**

W4B3   Paper falls short in translating these insights into actionable recommendations.    Rebuttal gives three examples.

QKiR  The reproducibility story in the main text omits detailed settings. Provided in rebuttal.

Denk Validity concerns due to prompts being manually constructed. Rebuttal mentioned a clinical neuroscientist reviewed the prompt templates.


Limitations due to UMAP. Rebuttal recomputed the circularity analysis entirely without UMAP.

Ablations: prompt sensitivity, alternative distances, non-medical tasks, generic behavior.   Authors  evaluated six additional medically oriented LLMs.

 Reviewer bgnx.   Only one medical-specific LLM is experimented with in this paper:  Authors have since ran 6 additional medical/biology-specific LLMs

**Reviewer Scores:**

QKiR  remain

Denk  remain or increase

W4B3  remain or increase.

bgnx will increase as mentioned in discussion.

---

### Decision · Program_Chairs · 2026-01-26

Accept (Poster)